# The Hidden Cost of Structured Generation in LLMs:
# Draft-Conditioned Constrained Decoding

Avinash Reddy [1]   Thayne T. Walker [2]   James S. Ide [2]   Amrit Singh Bedi [1]

## Abstract

Large language models (LLMs) are increasingly used to generate executable outputs, JSON objects, and API calls, where a single syntax error can make the output unusable. Constrained decoding enforces validity token-by-token via masking and renormalization, but it can distort generation when the model assigns low probability mass to valid continuations, pushing decoding toward locally valid yet semantically incorrect trajectories. We propose *Draft-Conditioned Constrained Decoding (DCCD)*, a simple two-step, training-free inference procedure that decouples semantic planning from structural enforcement: an unconstrained draft is generated first, and constrained decoding is then applied, conditioned on this draft, to guarantee validity. We analyze DCCD through a KL-projection view, showing that draft conditioning increases feasible mass and reduces the cumulative "projection tax" induced by hard constraints, with an optional best-of-$K$ draft selection. Across structured reasoning benchmarks, DCCD improves strict structured accuracy by up to +24 percentage points over standard constrained decoding (e.g., 15.2% to 39.0% on GSM8K with a 1B model), and enables smaller model pairs to match or exceed much larger constrained baselines, yielding substantial gains in parameter efficiency. We release code to reproduce all experiments at https://github.com/avinashreddydev/dccd.

## 1. Introduction

Large language models (LLMs) are increasingly deployed not just as chatbots but as components in software pipelines

[1]Department of Computer Science, University of Central Florida, FL, US [2]Lockheed Martin AI Center, CT, US. Correspondence to: Amrit Singh Bedi <amritbedi@ucf.edu>.

*Proceedings of the 43rd International Conference on Machine Learning*, Seoul, South Korea. PMLR 306, 2026. Copyright 2026 by the author(s).

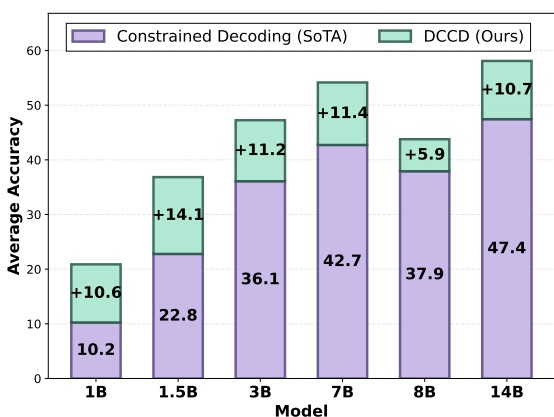

*Figure 1.* Our proposed approach, Draft-Conditioned Constrained Decoding (DCCD), yields consistent accuracy improvements over standard constrained decoding (state of the art) across model scales (1B–14B). Purple bars denote baseline constrained decoding accuracy, while green segments show the absolute accuracy gain from DCCD. These gains reflect improved response correctness and structure adherence at all model sizes.

that must produce machine-interpretable outputs. This shift is central to tool-augmented LLM systems and agentic workflows (e.g., Toolformer (Schick et al., 2023), ReAct (Yao et al., 2022), MRKL-style tool routers (Karpas et al., 2022)). In these settings, syntactic validity is non-negotiable, because a single missing brace in JSON, or an invalid SQL keyword, can cause downstream execution to fail. As a result, *structured generation*, which produces outputs that must satisfy hard constraints such as JSON schemas, grammars, or tool-call formats, has become a practical bottleneck for reliable LLM deployment (OpenAI, 2025).

**Key existing approaches.** A first class of methods relies on prompt-based format control (schema instructions, few-shot demonstrations, and reminders (Elnashar et al., 2025).) These can improve structural adherence, but they do not guarantee correctness and can still produce invalid outputs. A second, widely used approach is *constrained decoding* (CD), which enforces validity during generation by masking invalid tokens at every step. This includes lexically CD (Yin & Neubig, 2017; Xiao et al., 2016; Hokamp & Liu, 2017; Post & Vilar, 2018), finite-state CD (Park et al., 2025), and systems that integrate grammar engines (e.g., PICARD for text-to-SQL (Scholak et al., 2021)) as well as more recent

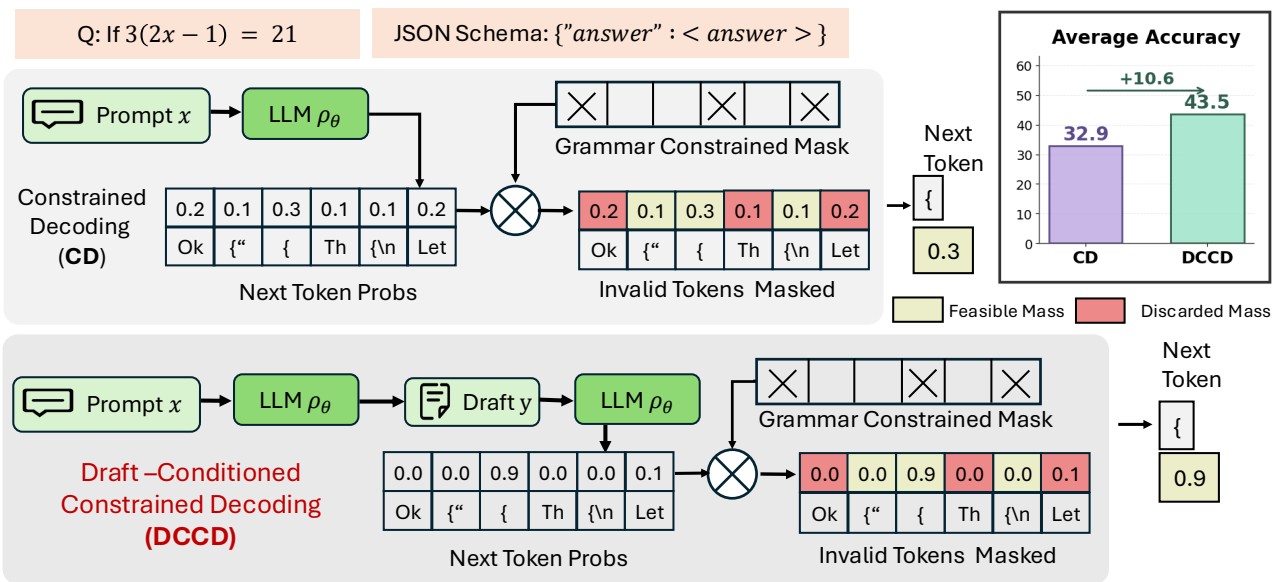

*Figure 2.* **Draft-Conditioned Constrained Decoding (DCCD) reduces distortion from structural constraints.** Given a prompt $x$ requiring a JSON-schema-conformant answer, standard constrained decoding (CD) applies a grammar-constrained mask directly to the base policy $\rho_\theta(\cdot \mid x)$, zeroing out invalid tokens and renormalizing the remainder. When the feasible mass is low, as for the first token here, where most probability sits on natural-language continuations like "Ok" or "Th", the renormalization distorts the distribution. **DCCD** instead first samples an unconstrained draft $y \sim \rho_\theta(\cdot \mid x)$ and conditions the next-token distribution on the draft, $\rho_\theta(\cdot \mid x, y)$, before applying the grammar mask. Conditioning on the draft concentrates probability mass on schema-compatible tokens (e.g., {}), so the feasible mass is already high, and the projection introduces minimal distortion. DCCD improves average accuracy by +10.6 points (32.9 $\rightarrow$ 43.5) over standard CD evaluated over 6 models and 4 datasets.

XGrammar (Dong et al., 2025) and Outlines (Willard & Louf, 2023). Constrained decoding guarantees that every emitted token preserves global validity, so the final output response always satisfies the structural constraint.

**Challenge of constrained decoding.** Despite guaranteeing validity, constrained decoding often reduces semantic correctness on reasoning-intensive tasks (Tam et al., 2024; Castillo, 2024; Schall & De Melo, 2025). The root cause is that constrained decoding is not a passive formatting filter: it *alters the model's distribution at every token* by masking invalid tokens and renormalizing the remaining probability over the valid ones. When strict formats force low-entropy syntax decisions (e.g., braces, quotes, commas, or field names), the model may place only a little probability on the valid options at a prefix, so renormalization becomes a large perturbation. Repeating such perturbations across many steps induces a *trajectory bias*: decoding is systematically pushed toward prefixes that are easier to keep valid, even when they correspond to an incorrect underlying solution.

**Our key insight.** The severity of this distortion is not intrinsic to the constraint alone, it depends on the *context* the model is conditioned on. If we can first supply a semantic plan that makes schema-consistent continuations likely, then the same hard constraints become far less distortive. This motivates a simple strategy: rather than forcing the

model to *reason inside* the constraint set, we first generate an unconstrained draft (capturing the semantic plan) and then apply constrained decoding *conditioned on the draft* to guarantee validity. This "draft-then-constrain" approach reduces constraint-induced distortion while preserving exact structural guarantees.

**Our solution.** Motivated by this view, we propose Draft-Conditioned Constrained Decoding (DCCD), a lightweight, training-free two-step inference procedure, illustrated in Figure 2. In Step 1, a model generates an *unconstrained draft* that captures the semantic plan or intermediate reasoning. In Step 2, we generate the final structured output using constrained decoding *conditioned on the draft*. Conditioning shifts probability mass toward schema-consistent continuations, so the subsequent constraint enforcement step is substantially less distortive. Because Step 2 primarily performs *structured realization* rather than open-ended reasoning, it can often be carried out by the same model or a smaller projector model, improving parameter efficiency. We summarize our contributions as follows.

● **Highlighting why constrained decoding fails.** We present a KL-projection perspective of constrained decoding and show that constraint-induced distortion is governed by the probability mass assigned to valid continuations (feasible mass), yielding a cumulative "projection tax" and

trajectory-dependent bias under hard constraints.

• **Draft-Conditioned Constrained Decoding (DCCD).** We introduce a training-free two-step inference algorithm that generates an unconstrained draft (semantic plan) and then performs constrained decoding conditioned on the draft, increasing feasible mass *before* enforcing hard constraints while retaining exact validity guarantees. We also show the test-time scaling effectiveness of the DCCD approach.

• **Empirical results.** Across multiple structural constraint types (JSON schemas, expression grammars, and prover-checked logical forms) and reasoning benchmarks (GSM8K, MATH500, GSM-Symbolic, and FOLIO/P-FOLIO), DCCD consistently improves *strict structured accuracy* (correct *and* valid) and enables parameter-efficient two-model compositions that outperform larger single-model constrained baselines. For example, under a strict JSON constraint on GSM8K, DCCD improves a 1B model from 15.24% to 39.0% strict accuracy, and improves a 1.5B model from 49.36% to 73.92%.

## 2. Related Works

The landscape of constrained decoding spans several algorithmic paradigms. Early work on lexically constrained decoding (Hokamp & Liu, 2017; Post & Vilar, 2018) enforced specific word or phrase inclusion in neural machine translation. This evolved into grammar-based approaches that integrate incremental parsing or automaton engines to enforce context-free grammars (Scholak et al., 2021; Park et al., 2025). Modern structured decoding systems (Willard & Louf, 2023; Dong et al., 2025; Suresh et al., 2025; Ugare et al., 2024a;b; OpenAI, 2024) extend these ideas to JSON schemas, type systems, and domain-specific languages, achieving efficient token-level filtering through optimized finite-state machines and parser integration. These systems are widely deployed in production LLM APIs (OpenAI, 2024; Anthropic, 2025) and form the backbone of tool-calling infrastructure.

Recent works (Tam et al., 2024; Castillo, 2024; Schall & De Melo, 2025) have documented performance degradation by 10-30% compared to unconstrained generation under hard structural constraints. Several approaches attempt to reduce this semantic distortion. Interleaved reasoning frameworks (Banerjee et al., 2025) allow models to alternate between free-form reasoning and structured generation, deferring structure until reasoning is complete. However, when structure is required from the first token, as in tool calls or API arguments, these methods reduce to standard constrained decoding, inheriting the same quality-validity tradeoff. Our work argues that the quality–validity trade-off is largely an artifact of how constraints are enforced at inference time. We show that one can preserve exact structural

guarantees while matching (or approaching) unconstrained task accuracy, without relaxing the constraints.

## 3. Problem Formulation

Let $x$ denote an input prompt and let $z_{1:T}$ denote an output token sequence of length $T$ over a vocabulary $\mathcal{V}$. A pretrained autoregressive language model $\pi_\theta$ (base model) induces the sequence distribution

$$\rho_\theta(z_{1:T} \mid x) \ = \ \prod_{t=1}^{T} \pi_\theta(z_t \mid h_t), \qquad h_t \triangleq (x, z_{<t}). \quad (1)$$

**Hard structural constraints.** We consider hard (non-negotiable) constraints that define a set of valid outputs. Formally, let $\mathcal{L}(x) \subseteq \mathcal{V}^\star$ denote the set of all sequences that satisfy the required structure for input $x$ (e.g., a JSON schema, a context-free grammar, a tool-call signature, or a proof/program syntax). A standard way to represent such constraints is via the valid-next-token set. For each state $h_t = (x, z_{<t})$, we define

$$A(h_t) \ \triangleq \ \{a \in \mathcal{V} : \exists \text{ a completion } z_{t+1:T} \text{ s.t.}$$
$$(z_{<t}, a, z_{t+1:T}) \in \mathcal{L}(x)\}. \quad (2)$$

Equivalently, one can define a prefix-dependent binary mask $m_t \in \{0, 1\}^{|\mathcal{V}|}$ with $m_t(a) = \mathbb{I}[a \in A(h_t)]$ for $a \in \mathcal{V}$. This abstraction covers grammar-constrained decoding, JSON-schema constrained generation, and executable-output formats used in semantic parsing and tool calling (e.g., lexically constrained decoding , finite-state constrained decoding (Willard & Louf, 2023; Dong et al., 2025; Suresh et al., 2025; Ugare et al., 2024a;b; OpenAI, 2024) , and incremental parsing constraints such as PICARD (Scholak et al., 2021).

**Objective.** The goal is to generate a response from the language model $\pi_\theta$ for a given $x$ that always satisfies the structure constraint while preserving task utility. Let $U(x, z)$ denote a task-specific utility (e.g., exact-match accuracy, execution success, or prover verification). We aim to produce a constrained distribution $q$ over sequences such that $q$ is supported on $\mathcal{L}(x)$ and attains high expected utility:

$$\max_{q} \ \mathbb{E}_{z \sim q}[U(x, z)] \quad \text{s.t.} \quad q(z \notin \mathcal{L}(x)) = 0. \quad (3)$$

Because $U(\cdot)$ is typically unavailable at inference time, most practical methods (e.g., constraint decoding) enforce constraints while attempting to stay close to the base model distribution.

### 3.1. Existing Approach: Constrained Decoding

The standard constrained decoding approaches (Hokamp & Liu, 2017; Post & Vilar, 2018; Willard & Louf, 2023;

Dong et al., 2025) enforces constraints during generation by masking invalid tokens and renormalizing at each step . Concretely, it defines the constrained per-step distribution

$$q(z_t \mid h_t) \;=\; \frac{\pi_\theta(z_t \mid h_t)\, \mathbb{I}[z_t \in A(h_t)]}{\alpha(h_t)}, \qquad (4)$$

where $\alpha(h_t) \triangleq \sum_{a \in A(h_t)} \pi_\theta(a \mid h_t)$, is the feasible mass at step $t$. We call $\alpha(h_t)$ the feasible mass because it is the total probability that the model assigns to feasible (i.e., constraint-valid) next tokens for a given $h_t$. Then, we sample $z_t \sim q(\cdot \mid h_t)$ (or greedily takes $\arg\max$ under $q$), guaranteeing that the final output lies in $\mathcal{L}(x)$.

The quantity $\alpha(h_t)$ is the feasible mass that the base model assigns to valid continuations at prefix $h_t$. Masking and renormalization introduce a per-step reverse-KL distortion

$$\mathrm{KL}(q(\cdot \mid h_t) \,\|\, \pi_\theta(\cdot \mid h_t)) \;=\; \log \frac{1}{\alpha(h_t)}. \qquad (5)$$

Thus, whenever $\alpha(h_t) \ll 1$, constrained decoding substantially reshapes the distribution over the remaining valid tokens. Further, aggregating across time yields a sequence-level projection tax. Let $\rho_q(z_{1:T} \mid x) \triangleq \prod_{t=1}^{T} q(z_t \mid h_t)$ denote the constrained autoregressive factorization induced by (4). Then the total distortion relative to the base model admits the identity

$$\mathrm{KL}(\rho_q(\cdot \mid x) \,\|\, \rho_\theta(\cdot \mid x)) = \mathbb{E}_{z \sim \rho_q(\cdot \mid x)} \left[ \sum_{t=1}^{T} \log \frac{1}{\alpha(h_t)} \right],$$

which shows that constrained decoding pays an additive projection tax: whenever $\alpha(h_t)$ is small for many steps, the cumulative KL distortion can become large.

**KL projection view.** Let $\Delta_{A(h_t)}$ denote the probability simplex supported on $A(h_t)$:

$$\Delta_{A(h_t)} \triangleq \{ p \in \Delta(\mathcal{V}) : \mathrm{supp}(p) \subseteq A(h_t) \}. \qquad (6)$$

Then the renormalized distribution in (4) is the unique solution to the reverse-KL projection problem

$$q(\cdot \mid h_t) \;=\; \arg\min_{p \in \Delta_{A(h_t)}} \mathrm{KL}(p \,\|\, \pi_\theta(\cdot \mid h_t)). \qquad (7)$$

That is, constrained decoding can be interpreted as repeatedly projecting $\pi_\theta(\cdot \mid h_t)$ onto the constraint set in KL geometry.

A subtle but important consequence of token-level renormalization is that the induced sequence distribution over valid strings is not restricted to $\mathcal{L}(x)$. For any valid $z \in \mathcal{L}(x)$, expanding Eq. (4) gives

$$\rho_q(z \mid x) = \prod_{t=1}^{T} \frac{\pi_\theta(z_t \mid h_t)}{\alpha(h_t)} = \frac{\rho_\theta(z \mid x)}{\prod_{t=1}^{T} \alpha(h_t)}. \qquad (8)$$

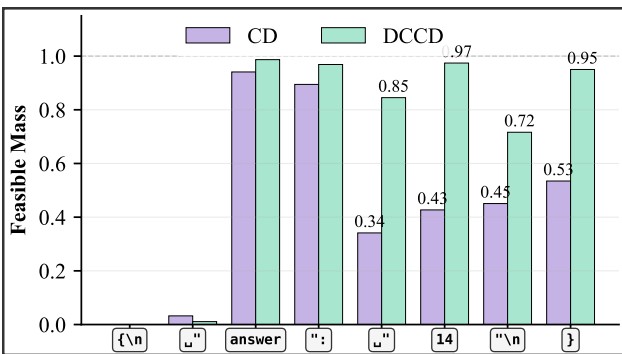

*Figure 3.* **Low feasible mass results in distortion.** Across tokens in a toy example, the feasible mass $\alpha(h_t)$ under constrained decoding is always $< 0.53$ and is near zero for early tokens. In this setting, the constraint admits only $\approx 1\%$ of the full vocabulary as feasible tokens, forcing strong renormalization and accumulating KL distortion.

Thus, even among valid sequences, constrained decoding reweights candidates by a prefix-dependent factor $\left( \prod_t \alpha(h_t) \right)^{-1}$. When feasible mass varies significantly across prefixes, this trajectory-dependent reweighting can steer decoding toward locally easy-to-project prefixes rather than globally correct solutions.

**A toy example.** Consider a task whose output must be a single-slot JSON object of the form $z = \{\texttt{"answer"}: a\}$, where $a \in \mathcal{V}_{\mathrm{ans}}$ is a single semantic token. The grammar forces a fixed sequence of formatting tokens $s_{1:m}$ (e.g., {, quotes, field name, colon, delimiters), followed by the semantic token $a$ at position $m+1$. Thus, most decoding steps are formatting steps, and at many positions the valid-next-token set is a singleton. If $A(h_t) = \{s_t\}$, constrained decoding deterministically emits $s_t$, and the renormalization factor is $\alpha(h_t) = \pi_\theta(s_t \mid h_t)$. When the base model would naturally respond in free-form text (e.g., "The answer is 14."), early schema tokens such as '{' can have very low probability, making $\alpha(h_t)$ small (see Figure 15 an Appendix) and the per-step distortion large. Because the model is autoregressive, forcing a sequence of such low-probability formatting tokens drives generation through unlikely pre-

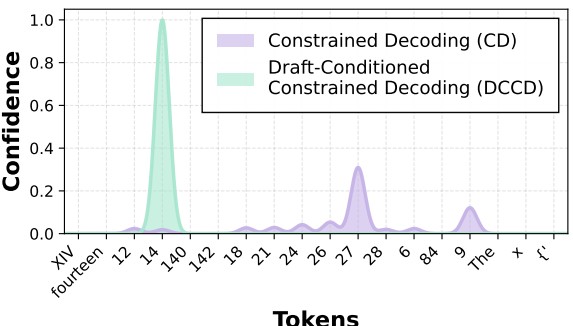

**Token-wise Distribution of Answer Token**

*Figure 4.* Token-wise confidence distribution for answer tokens in a single example. Constrained decoding spreads probability mass across multiple plausible answer tokens ("6", "27", "28", "6", "84", "9"), with the incorrect answer "27" receiving moderate confidence (0.46). DCCD shows a sharp, concentrated distribution with near-perfect confidence (1.0) on the correct token "14".

fixes, which can shift the model's downstream distribution at the semantic slot $a$.

**Key challenge.** In structured formats, low feasible mass often arises not because the model lacks the semantic solution, but because the schema enforces specific low-entropy tokens at specific times (quotes, delimiters, field names, operator symbols, etc.). As a result, $\alpha(h_t) \ll 1$ can occur repeatedly across a generation (see Figure 15), causing constraint enforcement to accumulate substantial distortion and induce trajectory-dependent bias. Empirically, this shows up as outputs that are perfectly well-formed yet semantically incorrect under strict constrained decoding (Koo et al., 2024). This points to an actionable remedy: reduce distortion by increasing feasible mass before enforcing constraints, i.e., steer the model toward constraint-consistent continuations first, and only then apply hard token-level masking to retain exact validity guarantees.

# 4. Draft-Conditioned Constrained Decoding

## 4.1. Key insight: feasible mass is context-dependent

Section 3 showed that the distortion induced by constrained decoding is governed by the feasible mass

$$\alpha(h_t) = \sum_{a \in A(h_t)} \pi_\theta(a \mid h_t), \qquad (9)$$

since $\text{KL} = \log \frac{1}{\alpha(h_t)}$ (cf. (5)). At first glance, this suggests that at inference time, the model parameters are fixed, and the valid set $A(h_t)$ is fixed by the schema/grammar. So what can we possibly change to make $\alpha(h_t)$ larger to reduce distortion? The key observation is that while $\pi_\theta$ is fixed, the conditional distribution is not, it depends on the conditioning context. In particular, if we append any auxiliary text $d$ (a

"draft", "plan", or intermediate representation) to the context, the model induces a different next-token distribution $\pi_\theta(a \mid h_t, d)$. This changes the feasible mass to

$$\alpha(h_t; d) \triangleq \sum_{a \in A(h_t)} \pi_\theta(a \mid h_t, d). \qquad (10)$$

Thus, even though $A(h_t)$ is fixed, we can increase the probability mass on valid tokens by choosing an appropriate auxiliary context $d$. Our method will instantiate $d$ as an unconstrained draft generated by a language model, and then enforce the hard constraint only after conditioning on this draft. Now consider adding an auxiliary context $d$ that contains the intended content (e.g., a draft reasoning trace that ends with the correct answer). Conditioning on $d$ changes the distribution over formatting tokens:

$$\pi_\theta(s_t \mid h_t, d) \gg \pi_\theta(s_t \mid h_t) \Rightarrow \alpha(h_t; d) \gg \alpha(h_t), \quad (11)$$

which reduces the projection tax at each forced formatting. As an example, an unconstrained response as auxiliary context can help improve the likelihood of a correct token, as shown in Figure 4. Crucially, it also makes the forced prefix $s_{1:m}$ in-distribution under the conditioned model, so the answer-slot distribution is better preserved.

## 4.2. Proposed Algorithm

Motivated by Eq. (10), we implement the auxiliary context $d$ as an unconstrained draft $y$. DCCD comprises of two autoregressive models: a draft model $p_{\text{draft}}$ for free-form planning and a projector model $p_{\text{proj}}$ for structure-constrained generation. They may be the same model ($p_{\text{draft}} = p_{\text{proj}}$), or $p_{\text{proj}}$ could be smaller since this stage primarily performs structured realization.

**Step 1: Draft generation.** We first sample a draft

$$y \sim p_{\text{draft}}(\cdot \mid x), \qquad (12)$$

where $y$ can be a plan, an outline, or free-form reasoning and is *not* required to satisfy the hard constraint.

**Step 2: Draft-conditioned constrained decoding.** We then generate the final structured output $z_{1:T}$ using constrained decoding conditioned on $(x, y)$. We define

$$p_2(z_t \mid \tilde{h}_t), \qquad \tilde{h}_t \triangleq (x, y, z_{<t}), \qquad (13)$$

where $p_2$ is instantiated by $p_{\text{proj}}$ (which is $\pi_\theta$ in our case) with the draft $y$ included in context. The hard constraint still applies to the final output prefix $h_t = (x, z_{<t})$. Therefore, the valid-next-token set remains $A(h_t)$ from Section 3. We enforce validity by masking and renormalizing under the conditioned distribution:

$$\tilde{q}(z_t \mid \tilde{h}_t) = \frac{p_2(z_t \mid \tilde{h}_t) \, \mathbb{I}[z_t \in A(h_t)]}{\tilde{\alpha}(\tilde{h}_t)}, \qquad (14)$$

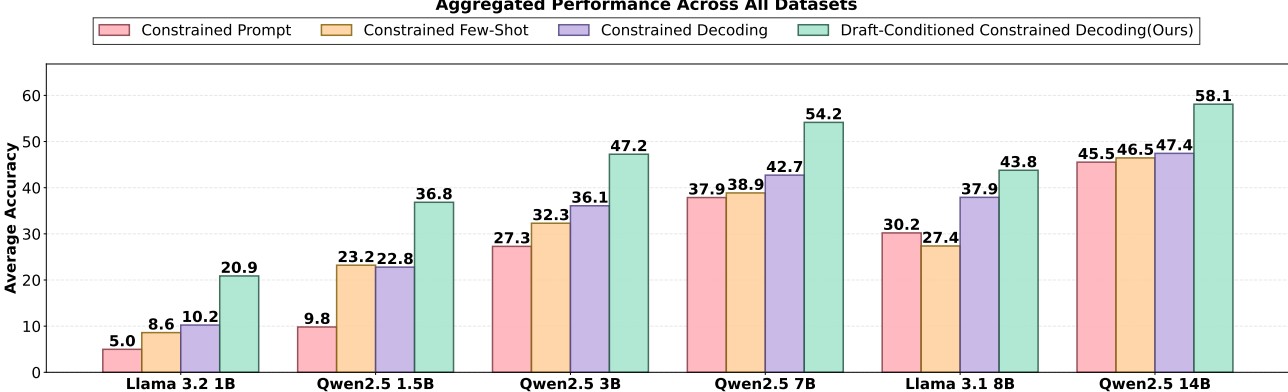

*Figure 5.* **Average performance comparison across all evaluation datasets** (GSM8K, GSM Symbolic, Math500, and FOLIO). We compare prompting-based baselines (CP, CF), grammar-based constrained decoding (CD), and our Draft-Conditioned Constrained Decoding (DCCD). Across all model scales, DCCD achieves the best aggregated performance, with the largest relative gains for smaller models where hard constraints induce the strongest projection distortion (e.g., 1B: 10.2% →20.9%). **Takeaway:** Conditioning on an unconstrained draft before enforcing hard constraints yields consistent, model-agnostic improvements in strict structured generation.

with draft-conditioned feasible mass $\tilde{\alpha}(\tilde{h}_t) \triangleq \sum_{a \in A(h_t)} p_2(a \mid \tilde{h}_t)$. We decode by sampling $z_t \sim \tilde{q}(\cdot \mid \tilde{h}_t)$ (or greedy decoding).

**Key connection to the challenge in Section 3.** Section 3 identified low feasible mass $\alpha(h_t)$ as the driver of projection tax and semantic distortion. DCCD targets this directly: it seeks drafts $y$ such that $\tilde{\alpha}(\tilde{h}_t) \gg \alpha(h_t)$ along the realized trajectory. In practice, the draft makes structural tokens (quotes, braces, delimiters, field names) much more probable, reducing repeated forced "surprises" and preserving the model's semantic preferences for the content tokens.

---

**Algorithm 1** Draft-Conditioned Constrained Decoding

---

**Require:** Prompt $x$; draft model $p_{\text{draft}}$; projector model $p_{\text{proj}}$; constraint oracle $A(\cdot)$; max length $T$; number of drafts $K$
1: Sample drafts $y^{(1)}, \ldots, y^{(K)} \sim p_{\text{draft}}(\cdot \mid x)$
2: **for** $k = 1$ to $K$ **do**
3:     $z_{<1}^{(k)} \leftarrow \emptyset, \quad S^{(k)} \leftarrow 0$
4:     **for** $t = 1$ to $T$ **do**
5:         Compute $p_2(\cdot \mid x, y^{(k)}, z_{<t}^{(k)})$ using $p_{\text{proj}}$
6:         $\tilde{\alpha}_t^{(k)} \leftarrow \sum_{a \in A(x, z_{<t}^{(k)})} p_2(a \mid x, y^{(k)}, z_{<t}^{(k)})$
7:         $S^{(k)} \leftarrow S^{(k)} + \log(\tilde{\alpha}_t^{(k)})$
8:         Form $\tilde{q}(\cdot) \propto p_2(\cdot) \odot \mathbb{I}[\cdot \in A(x, z_{<t}^{(k)})]$
9:         Sample (or greedily select) $z_t^{(k)} \sim \tilde{q}(\cdot)$
10:     **end for**
11: **end for**
12: Choose $k^\star \in \arg\max_k S^{(k)}$ (or $k^\star = 1$ when $K{=}1$)
13: **return** $z_{1:T}^{(k^\star)}$

---

We summarize the proposed steps in Algorithm 1. Algorithm 1 presents DCCD in a general form that also subsumes

several practical extensions. In particular, the algorithm allows generating multiple unconstrained drafts in parallel ($K > 1$) and selecting the final output via a late-selection criterion. In our instantiation, we score each candidate using the cumulative log feasible mass incurred during constrained decoding, which directly reflects the amount of constraint-induced distortion. However, the framework itself is agnostic to the specific selection rule: alternative criteria such as total log-likelihood under the constrained model, external verifier scores, task-specific judges, or majority voting across valid realizations can be substituted without changing the core procedure. When $K = 1$, DCCD reduces to a simple two-step draft-then-constrain decoding scheme.

## 5. Experiments

We evaluate our proposed approach (DCCD) on structured generation tasks that require both (i) semantic correctness and (ii) exact structural validity. Our experiments are designed to answer three questions:

**Q1** (Effectiveness). Does DCCD improve strict structured accuracy compared to prompting-based methods and standard constrained decoding?

**Q2** (Efficiency). Because DCCD composes two models, can it achieve better parameter/cost efficiency than single-model constrained decoding?

**Q3** (Test-time scaling). How does DCCD scale with additional test-time compute (e.g., sampling $n$ candidates and voting/selecting)?

Before presenting the results, we first describe the experimental setup, including the datasets, structural constraints, models, baselines, and evaluation metrics.

## 5.1. Experimental Setup

**Datasets and structural constraints.** We evaluate on GSM8K (Cobbe et al., 2021), MATH500 (Hendrycks et al., 2021), GSM-Symbolic (Mirzadeh et al., 2024), and FO-LIO (Han et al., 2024), spanning numerical math, symbolic math, and first-order logic formalization. For each dataset, we enforce a strict, machine-checkable output format (JSON schemas, expression grammar, or FOL grammar), with detailed examples of structures in Appendix F. (see Appendix B for more details).

**Models and baselines.** We test instruction-tuned models ranging from 1B to 14B parameters (Table 1). We compare against: (i) Constrained Prompting (CP) and Constrained Few-Shot (CF) (prompt-only format enforcement), and (ii) Constrained Decoding (CD) using XGrammar (Dong et al., 2025), which guarantees structural validity by masking invalid tokens.

*Table 1.* Models used.

| Size | Model |
|------|-------|
| 14B | Qwen2.5-14B |
| 8B | Llama-3.1-8B |
| 7B | Qwen2.5-7B |
| 3B | Qwen2.5-3B |
| 1.5B | Qwen2.5-1.5B |
| 1B | Llama-3.2-1B |

**Evaluation metric.** Our evaluation framework assesses two aspects of the response: (i) answer correctness by comparing the model's final answer against the ground truth, and (ii) structural compliance by verifying that the output strictly adheres to the specified format constraints (JSON schema for mathematical tasks, logical formalism for FO-LIO). Then, a response is marked as successful only when *both* conditions are satisfied. This joint evaluation captures the core challenge of structured generation: maintaining reasoning quality while satisfying hard constraints.

## 5.2. Main Results

**A1: DCCD improves strict structured accuracy across model scales and constraint types.** Figure 5 summarizes the main result: when we average strict accuracy across all benchmarks, DCCD consistently outperforms constrained prompting (CP), constrained few-shot (CF), and standard constrained decoding (CD) for every model scale from 1B to 14B. Table 2 then breaks this aggregate view into per-dataset results, showing that the gains hold across heterogeneous constraint types (JSON schemas, expression grammars, and prover-checked logical forms). Overall, DCCD delivers the best performance in all model–dataset configurations, with particularly large improvements for smaller models, which is consistent with our projection-tax view that draft conditioning increases feasible mass before masking and thereby reduces constraint-induced distortion.

**Takeaway:** DCCD is a training-free decoding strategy that reliably boosts strict structured generation accuracy across models and constraint types, with particularly large benefits in the low-parameter regime where constrained decoding is most distortive.

*Table 2.* Comparison of constrained decoding strategies across datasets. CP: Constrained Prompting, CF: Constrained Few Shot, **CD**: Constrained Decoding, DCCD: Draft Conditioned Constrained Decoding.

| Model Size | Algorithm | Datasets | | | |
|---|---|---|---|---|---|
| | | GSM8K | GSM Symbolic | Math500 | FOLIO |
| **1B** | CP | 7.51 | 6.00 | 6.40 | 0.00 |
| | CF | 13.80 | 9.00 | 11.60 | 0.00 |
| | CD | 15.24 | 0.00 | 6.00 | 19.70 |
| | DCCD (ours) | **39.04** | **9.00** | **19.80** | **21.67** |
| **1.5B** | CP | 13.27 | 11.00 | 15.00 | 0.00 |
| | CF | 48.22 | 23.00 | 21.60 | 0.00 |
| | CD | 49.36 | 12.00 | 15.00 | 14.78 |
| | DCCD (ours) | **73.92** | **23.00** | **38.20** | **18.23** |
| **3B** | CP | 59.14 | 19.00 | 30.00 | 1.00 |
| | CF | 71.80 | 25.00 | 32.40 | 0.00 |
| | CD | 73.24 | 17.00 | 33.40 | 20.69 |
| | DCCD (ours) | **84.53** | **36.00** | **46.80** | **21.67** |
| **7B** | CP | 80.06 | 31.00 | 40.40 | 0.00 |
| | CF | 82.26 | 29.00 | 44.20 | 0.00 |
| | CD | 81.58 | 26.00 | 43.60 | 19.70 |
| | DCCD (ours) | **91.28** | **41.00** | **52.80** | **31.53** |
| **8B** | CP | 76.80 | 17.00 | 27.00 | 0.00 |
| | CF | 70.20 | 14.00 | 24.80 | 0.49 |
| | CD | 80.89 | 19.00 | 28.60 | 23.15 |
| | DCCD (ours) | **83.02** | **30.00** | **35.00** | **27.09** |
| **14B** | CP | 91.13 | 44.00 | 47.00 | 0.00 |
| | CF | 90.52 | 49.00 | 45.80 | 0.49 |
| | CD | 86.43 | 37.00 | 47.60 | 18.72 |
| | DCCD (ours) | **95.15** | **53.00** | **58.60** | **25.62** |

**A2: DCCD enables parameter-efficient model composition.** Figure 6 compares accuracy per billion parameters across all four benchmarks, highlighting how effectively each method uses model capacity. Across GSM8K, MATH500, GSM-Symbolic, and FOLIO, DCCD consistently achieves substantially higher accuracy per parameter than single-model constrained decoding, often by large margins. The gains are especially pronounced in low- to mid-capacity regimes: for example, on MATH500, a $1.5B+1.5B$ DCCD composition (3B total) achieves 12.7 accuracy per billion parameters, compared to 3.6 for an 8B model using constrained decoding, which is a 253% efficiency improvement. Similarly, on GSM8K, a $7B + 1.5B$ composition (8.5B total) outperforms a single 14B model (10.7 vs. 6.1 accuracy per billion), despite using 39% fewer parameters. This pattern holds consistently across tasks and scales, indicating that DCCD does not only shift performance, but fundamentally improves how parameters are utilized.

**Takeaway.** DCCD enables smaller, cheaper model pairs to match or exceed much larger constrained baselines, demonstrating that separating reasoning from formatting yields substantial gains in parameter efficiency.

**A3: DCCD scales better with test-time sampling than constrained decoding.** We evaluate test-time scaling by sampling $n \in \{1, 3, 5, 7, 9, 11, 13\}$ candidates and applying majority vote at inference time. For constrained decoding (CD), we vote over $n$ independently generated *structured* outputs. For DCCD, we vote over $n$ unconstrained

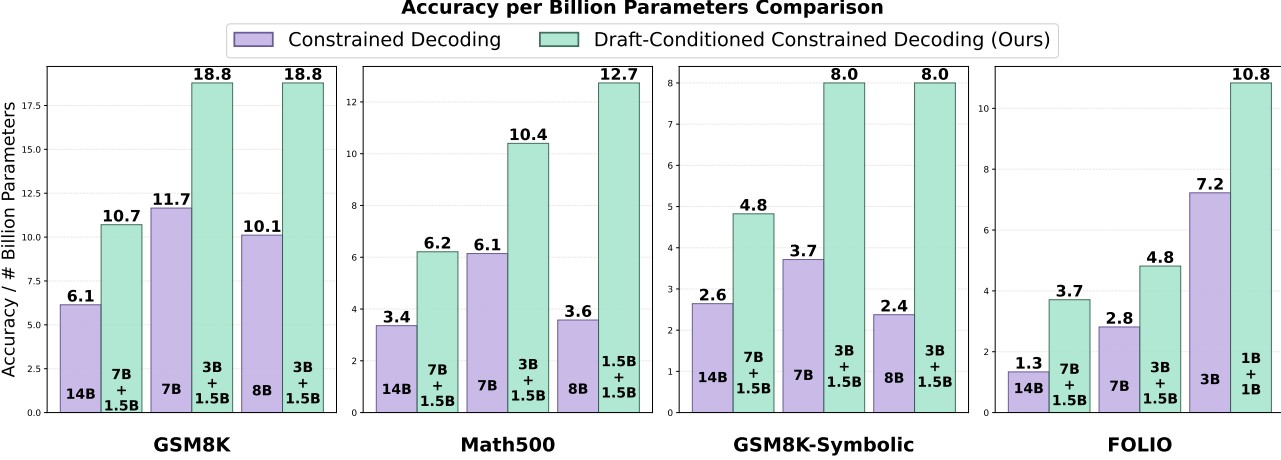

*Figure 6.* **Parameter efficiency (accuracy per billion parameters).** For each dataset, we report strict structured accuracy normalized by total parameters for single-model CD and parameter-matched DCCD compositions. DCCD consistently achieves higher accuracy per parameter, with the largest gains in low-capacity regimes.

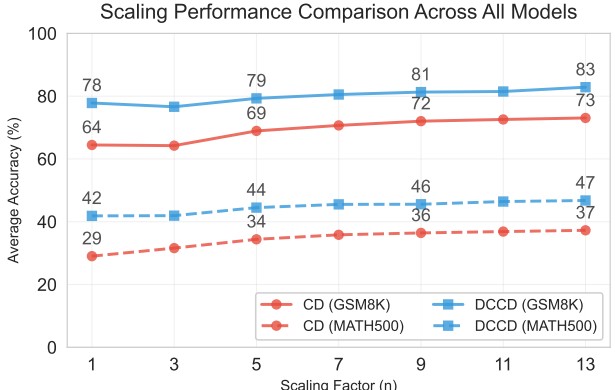

*Figure 7.* **Test-time scaling comparison** across GSM8K and MATH500 (averaged over six models, 1.5B-14B parameters). Solid lines: GSM8K; dashed lines: MATH500. Draft-Conditioned Constrained Decoding (blue) shows superior scaling versus Constrained Decoding (red), with widening performance gaps as n increases from 1 to 13.

*drafts* (Stage 1) and then run a single constrained projection (Stage 2) on the selected draft. Figure 7 shows that DCCD benefits more from additional test-time compute on both benchmarks. On GSM8K, DCCD improves from 78% at $n{=}1$ to 83% at $n{=}13$, while CD improves from 64% to 73%. On MATH500, DCCD improves from 42% to 47% whereas CD improves from 29% to 37%, and the performance gap remains large throughout. In both cases, gains saturate beyond moderate $n$ (around $n \approx 7$), suggesting diminishing returns once the best drafts/solutions are already sampled.

**Takeaway:** Allocating test-time compute to sampling diverse *drafts* (semantic plans) is more effective than repeatedly sampling under hard constraints, and DCCD converts this additional compute into higher structured accuracy.

## 5.3. Additional Insights

**Is constrained decoding the cause of accuracy degradation?** A natural concern is whether the observed accuracy drop comes from applying CD itself, or merely from mentioning the structural constraint in the prompt. To isolate these two effects, we evaluate three settings on MATH500 under *answer-correctness*: a *plain prompt without CD* as the baseline, a *structural prompt without CD* that explicitly describes the JSON schema in the prompt, and a *structural prompt with CD* that applies grammar masking at decode time on top of the same structural prompt. Comparing the first two settings isolates the effect of prompt-level structural awareness, while comparing the latter two isolates the effect of decode-time masking. We report the results in Table 3. As shown in Table 3, across all six model sizes, moving

| Model | Plain prompt, without CD | Structural prompt, without CD | Structural prompt, with CD |
|---|---|---|---|
| 1B | 24.6 | 20.6 (−4.0) | 6.0 (−18.6) |
| 1.5B | 45.6 | 41.4 (−4.2) | 15.0 (−30.6) |
| 3B | 54.2 | 50.3 (−3.9) | 33.4 (−20.8) |
| 7B | 71.4 | 71.2 (−0.2) | 43.6 (−27.8) |
| 8B | 39.2 | 38.0 (−1.2) | 28.6 (−10.6) |
| 14B | 76.0 | 70.8 (−5.2) | 47.6 (−28.4) |

*Table 3.* Prompt-vs-CD decomposition on MATH500 (answer accuracy). Parenthesized values are drops relative to the plain-prompt, without-CD baseline.

from the plain prompt to the structural prompt (without CD) costs at most $5.2$ points and as little as $0.2$, indicating that mentioning the structural constraint in the prompt has only a marginal effect on answer accuracy. In contrast, adding CD on top of the structural prompt costs an additional $10$–$30$ points in every setting, with the largest drops on the smaller

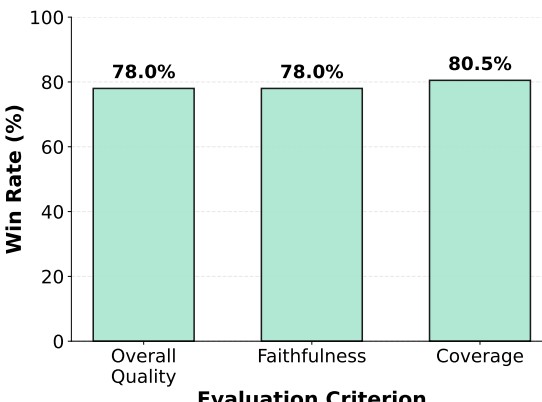

*Figure 8.* Win rate comparison between DCCD and CD on non-verifiable summarization tasks. LLM as a Judge assessed 256-token TL;DR summaries across three criteria: overall quality, faithfulness, and coverage. DCCD consistently outperforms CD with approximately 78–80.5% win rate across all evaluation dimensions, demonstrating the effectiveness of staged inference for reasoning-intensive generation tasks.

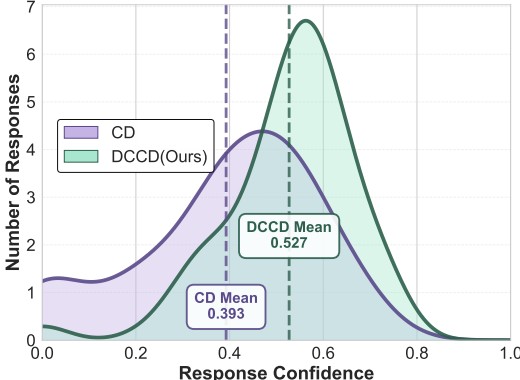

*Figure 9.* Distribution of response confidence scores for Llama 3.2 3B Instruct on GSM8K. The histogram compares the confidence distributions between standard Constrained Decoding (CD) using $q(y|x)$ and Draft-Conditioned Constrained Decoding (DCCD) using joint probability $p_{\text{draft}}(d|x) \cdot p_2(y|x, d)$. DCCD demonstrates a significant rightward shift with a mean confidence of 0.527 compared to CD's 0.393, indicating 39% higher confidence in generated responses.

models (e.g., $-30.6$ for the 1.5B model and $-28.4$ for the 14B model). This asymmetry shows that the degradation is caused primarily by *decode-time renormalization under low feasible mass*, rather than by the structural framing of the prompt, which is consistent with the mechanism analyzed in Section 4 and the KL-drift visualized in Figure 15.

**Response-Level Confidence.** We compared the response-level confidence of Llama 3.2 3B Instruct on the GSM8K dataset. Figure 9 shows the distribution of response probability confidence scores. We evaluated the final response confidence of CD as $q(y|x)$ and the joint probability of the draft and draft-conditioned final response in DCCD as $p_{\text{draft}}(d|x) \cdot p_2(y|x, d)$. DCCD elicited responses with 39% higher confidence compared to CD, which directly contributed to improved strict accuracy on the dataset.

**Non-Verifiable Tasks.** We designed an additional experiment to assess the robustness of DCCD on tasks without ground-truth verification. The objective was to generate 256-token TL;DR summaries across various topics (see Appendix E for details). We evaluated DCCD and CD responses using win rate percentage across three criteria: overall quality, faithfulness, and coverage. Figure 8 demonstrates that DCCD achieves approximately 80% win rate over CD across all evaluation categories. This finding supports our hypothesis that for reasoning-intensive tasks, a two stage inference procedure is preferable to direct generation.

## 6. Conclusion

We studied structured generation under hard constraints, where outputs must be both semantically correct and strictly valid (e.g., JSON schemas, expression grammars, and prover-checked logical forms). We showed that standard constrained decoding can be interpreted as repeated reverse-KL projections onto a prefix-valid set, and that its quality degradation is driven by low probability mass on valid continuations, incurring a cumulative "projection tax" and trajectory bias. Motivated by this view, we proposed Draft-Conditioned Constrained Decoding (DCCD), a training-free two-stage inference procedure that first generates an unconstrained draft (semantic plan) and then performs constrained decoding conditioned on the draft, increasing valid mass *before* projection while retaining exact validity guarantees. Across GSM8K variants, MATH500, and FOLIO/Prover9 and model scales from 1B to 14B, DCCD consistently improves strict structured accuracy, enables parameter-efficient model composition, and converts additional test-time compute into larger gains than standard constrained decoding.

## Acknowledgments

This work was sponsored and supported by Lockheed Martin.

## Impact Statement

This paper presents work whose goal is to advance the field of machine learning by improving the reliability of structured generation in large language models. While the proposed method may support more robust deployment of LLMs in software and tool-based systems, we do not foresee any new or unique societal risks beyond those already associated with large language models. We believe no additional ethical considerations are required beyond existing best practices for responsible AI deployment.

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

# Appendix

## Contents

## A. Why a larger feasible mass can improve accuracy

The toy example builds intuition, but we can also give a simple *distributional stability* justification.

Let $U(x,z) \in [0,1]$ be any bounded utility, and define the *validity-gated* utility

$$\bar{U}(x,z) \triangleq U(x,z)\,\mathbb{I}[z \in \mathcal{L}(x)]. \tag{15}$$

This makes invalid strings automatically receive zero utility, matching strict evaluation.

For any two sequence distributions $P$ and $Q$ over $\mathcal{V}^\star$,

$$\left|\mathbb{E}_{z \sim P}[\bar{U}(x,z)] - \mathbb{E}_{z \sim Q}[\bar{U}(x,z)]\right| \ \leq \ \mathrm{TV}(P,Q) \ \leq \ \sqrt{\tfrac{1}{2}\mathrm{KL}(P\|Q)}, \tag{16}$$

where the second inequality is Pinsker's inequality. Therefore, whenever constrained decoding yields a distribution $P$ that is *close in KL* to a reference distribution $Q$ that already assigns high utility to valid outputs, the constrained procedure cannot lose too much utility.

In our setting, standard constrained decoding defines $P = \rho_q(\cdot \mid x)$ and $Q = \rho_\theta(\cdot \mid x)$, and Section 3 shows

$$\mathrm{KL}(\rho_q(\cdot \mid x) \,\|\, \rho_\theta(\cdot \mid x)) = \mathbb{E}_{z \sim \rho_q(\cdot|x)}\left[\sum_{t=1}^{T} \log \frac{1}{\alpha(h_t)}\right]. \tag{17}$$

Thus, increasing feasible mass $\alpha(h_t)$ (especially on prefixes actually visited during decoding) directly reduces the KL term, which tightens the worst-case utility degradation bound in Eq. (16). This provides a principled reason that "making valid tokens higher-likelihood" can translate into higher strict accuracy: it reduces the amount by which the constraint mechanism can perturb the model away from its high-utility behavior.

## B. Additional details of experiments

**Datasets** We evaluate on four datasets spanning numerical mathematical reasoning, symbolic mathematical reasoning, and logical deduction:

1. **Numerical Math** We evaluate on three math datasets: **GSM8K** (Cobbe et al., 2021), comprising grade school math word problems requiring multi-step arithmetic reasoning; **MATH500** (Hendrycks et al., 2021), a subset of 500 problems from the MATH benchmark covering algebra, geometry, and number theory.
2. **Symbolic Math. GSM-Symbolic** (Mirzadeh et al., 2024), a symbolic variant of GSM8K designed to test genuine mathematical reasoning rather than pattern matching.
3. **Logical Reasoning. FOLIO** (Han et al., 2024), a first-order logic reasoning dataset requiring structured logical formalization. Given natural language premises and a conclusion, models must produce a formal representation. Outputs are verified using the Prover9 theorem, and accuracy is measured by whether the formalized proof matches the ground truth conclusion.

**Baselines** We compare DCCD three established approaches for structured generation, which fall into two categories based on their primary strengths:

The below methods prioritize generating correct answers but struggle with strict format adherence:
1. **Constrained Prompting(CP)** : System prompts are carefully engineered to specify the required output structure, including explicit format constraints and examples of valid outputs.
2. **Constrained Few-Shot (CF)**: In addition to format specifications, we provide $k = 3$ in-context examples that demonstrate the expected output structure, following standard few-shot prompting practices.

This below method guarantees format compliance but often sacrifices answer quality:
3. **Constrained Decoding (CD)**: We employ grammar-based constrained decoding using XGrammar (Dong et al., 2025) integrated with vLLM. At each decoding step, a token mask is constructed from the grammar specification, restricting sampling exclusively to syntactically valid tokens. This approach guarantees syntactic correctness but directly intervenes in the generation process, potentially limiting the model's reasoning capabilities.

## C. Probability of the Token Positions

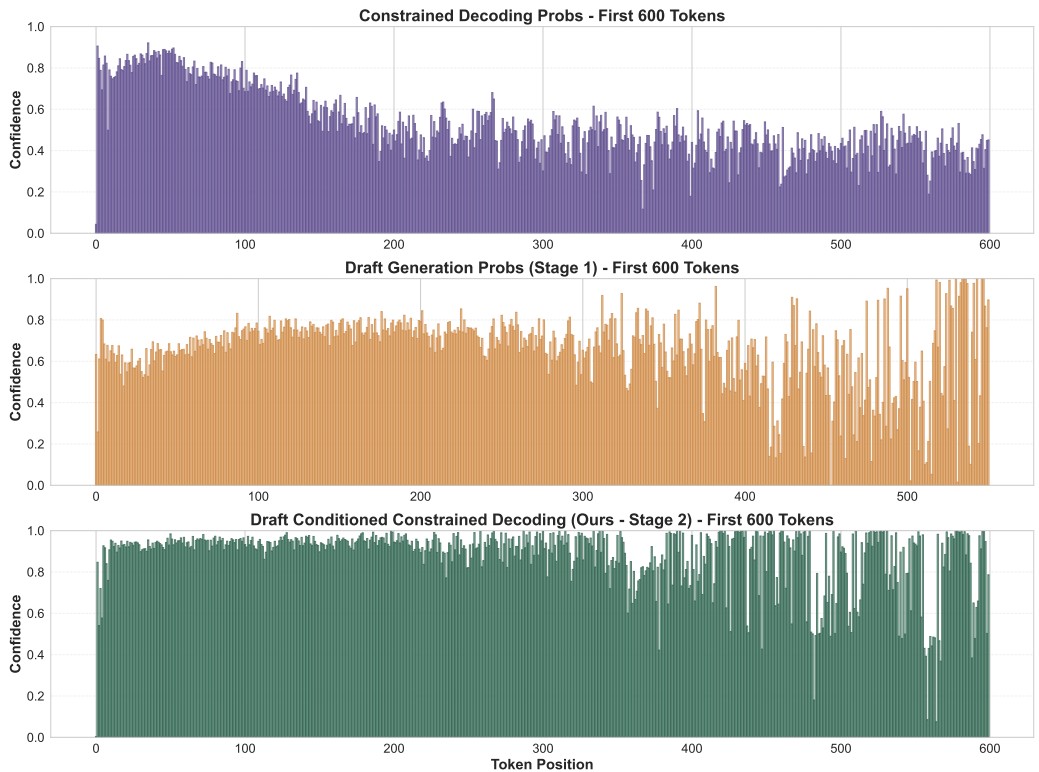

*Figure 10.* Token-wise probability distributions across the first 600 token positions for different decoding strategies. Constrained Decoding (top) shows degraded confidence over longer sequences, with probabilities declining from 0.8 initially to 0.35 in later positions. Draft Generation (middle) maintains consistent moderate probabilities throughout without structural constraints. Draft-Conditioned Constrained Decoding (bottom, ours) sustains high confidence across all positions, demonstrating that our Draft-Conditioned Constrained Decoding approach successfully maintains model confidence while enforcing structural constraints.

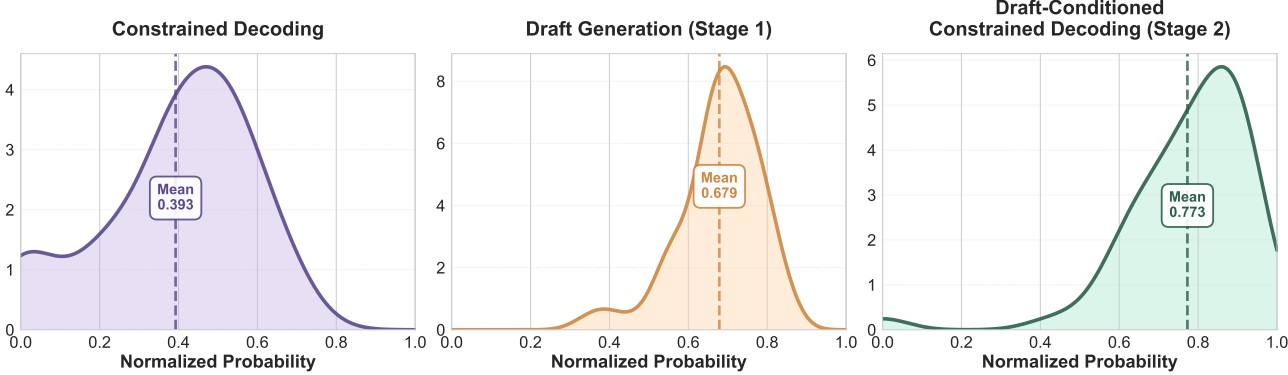

*Figure 11.* Probability distribution analysis across decoding strategies. (a) Normalized token-level probability distributions show that Draft-Conditioned Constrained Decoding (DCCD, Stage 2) achieves significantly higher mean probability (0.527) compared to Constrained Decoding (CD, 0.393), indicating improved model confidence. Draft Generation (Stage 1) serves as the intermediate unconstrained reasoning step. (b) Comparison between CD and DCCD Joint Probability (product of Stage 1 and Stage 2). Despite combining two probabilistic stages, the DCCD approach maintains substantially higher mean probability (0.527 vs 0.393), demonstrating that separating reasoning from formatting preserves model confidence while achieving structural constraints.

## D. Test Time Scaling: Detailed Results

Figure 12 presents the complete test-time scaling results for all six models on both GSM8K and MATH500 benchmarks. These detailed breakdowns complement the aggregated results shown in Figure 7 of the main paper.

Across all model sizes, we observe consistent patterns: 1) Draft-Conditioned Constrained Decoding(DCCD)(blue) consistently outperforms Constrained Decoding (red) at every scaling factor. 2) Smaller models (Llama 3.2 1B, Qwen2.5 1.5B) show larger absolute performance gaps, with Two-Stage Decoding maintaining significantly higher accuracy even at n=1. 3) Larger models (Qwen2.5 14B, Llama 3.1 8B, Qwen2.5 7B) demonstrate strong baseline performance with DCCD , approaching or exceeding 90% on GSM8K. 4) On the more challenging MATH500 benchmark, the scaling behavior is more gradual, with performance improvements continuing through n=13, though at diminishing rates. 5) Constrained Decoding exhibits more pronounced degradation on smaller models, particularly evident in the Llama 3.2 1B results where accuracy remains below 15% across all scaling factors on GSM8K.

The individual model trajectories reveal that the separation of reasoning from formatting provides consistent advantages across the entire model size spectrum, validating our approach's robustness and general applicability.

## E. Non-Verifiable Experimental Design and Evaluation

### E.1. Experimental Setup

We conduct a non-verifiable evaluation comparing CD against DCCD across diverse prompt categories under a strict 256-token budget constraint. Without ground-truth labels, we employ LLM-as-judge pairwise comparisons to assess relative quality. The model receives explicit instructions to generate content within a hard token budget ($\leq$256 tokens) while maintaining accuracy, coverage, and coherence. The prompt targets 95–100% budget utilization with sentence-boundary termination. Our proposed DCCD approach separates content generation from constraint satisfaction:

1. **Draft Generation:** Generate a comprehensive, unconstrained answer focusing on accuracy, coverage, and reasoning. The model is informed of the downstream budget to ensure sufficient detail for compression.

2. **Constrained Compression:** Compress the draft into a faithful summary within the hard budget, prioritizing coverage of core claims, key points, and critical numbers while avoiding fabrications.

*Table 4.* Evaluation prompt categories spanning diverse knowledge domains

| General Knowledge | Business & Economics |
|---|---|
| Science & Technology | AI & Computer Science |
| Philosophy & Psychology | Environment & Sustainability |
| Culture, Media & Design | Society, Law & Policy |
| Health, Medicine & Biology | Education & Productivity |

### E.2. Evaluation Methodology

E.2.1. LLM-AS-JUDGE FRAMEWORK

We employ **Qwen2.5-14B-Instruct** as an impartial evaluator for pairwise comparisons across three criteria: overall quality, coverage, and faithfulness. The judge produces structured XML output with explicit reasoning and a categorical decision (A/B/TIE).

---

**Judge System Prompt**

You are an impartial expert judge comparing two AI assistant answers. You MUST pick a single winner unless they are truly indistinguishable. Return XML with `<reasoning>...</reasoning>` and `<winner>A</winner>` or `<winner>B</winner>` or `<winner>TIE</winner>`. Do not include any other text outside the XML tags.

---

---

**Judge User Prompt Template**

Compare two candidate answers to the same prompt. Decide which is BETTER under the given criterion, considering the hard constraint budget of {budget} tokens.
**PROMPT:** {prompt}
**CRITERION:** {criterion_explanation}
**GENERAL RULES:**

- Faithfulness: no invented facts; consistency with prompt

- Coverage: include essential points to satisfy the prompt

- Quality: clear, coherent, well-organized

- Constraint sensitivity: maintain essential information within budget

**CANDIDATE A:** {answer_a}
**CANDIDATE B:** {answer_b}
**OUTPUT:** `<reasoning>...</reasoning>` `<winner>A/B/TIE</winner>`

---

**Evaluation Criteria**

**Overall Quality:** Prioritize correct, faithful, well-covered content that fits the budget and retains essential information.
**Coverage:** Prioritize answers that preserve essential points within budget while remaining clear and concise.
**Faithfulness:** Prioritize answers that avoid fabrications and remain faithful to the prompt while fitting the budget.

---

## F. Dataset Examples

This section provides complete examples from each benchmark dataset used in our evaluation, illustrating both the natural language questions and their corresponding structured outputs.

### F.1. GSM8K: Grade School Math

**GSM8K Example**

**Question.**

*Janet's ducks lay 16 eggs per day. She eats three for breakfast every morning and bakes muffins for her friends every day with four. She sells the remainder at the farmers' market daily for $2 per fresh duck egg. How much in dollars does she make every day at the farmers' market?*

**Structured output.**

```
{
  "steps": [
    "Calculate total eggs laid per day: 16",
    "Calculate eggs consumed daily: 3 + 4 = 7",
    "Calculate eggs remaining to sell: 16 - 7 = 9",
    "Calculate revenue at $2 per egg: 9 * 2 = 18"
  ],
  "answer": "18"
}
```

## F.2. GSM-Symbolic: Symbolic Mathematical Reasoning

> **GSM-Symbolic Example**
>
> **Question.**
>
> *{name} makes {drink} using teaspoons of sugar and cups of water in the ratio of {m}:{n}. If she used a total of {x} teaspoons of sugar and cups of water, calculate the number of teaspoonfuls of sugar she used.*
>
> **Structured output.**
>
> ```
> {
>     "answer": "<<(m*x)//(m+n)>>"
> }
> ```

## F.3. FOLIO: First-Order Logic Reasoning

> **FOLIO Example**
>
> **Question.**
>
> *All people who regularly drink coffee are dependent on caffeine. People either regularly drink coffee or joke about being addicted to caffeine. No one who jokes about being addicted to caffeine is unaware that caffeine is a drug. Rina is either a student and unaware that caffeine is a drug, or neither a student nor unaware that caffeine is a drug.*
>
> *Is the following statement true, false, or uncertain? Rina is either a person who jokes about being addicted to caffeine or is unaware that caffeine is a drug.*
>
> **Structured output.**
>
> ```
> Predicates:
> Dependent(x) ::: x is dependent on caffeine
> Drinks(x) ::: x regularly drinks coffee
> Jokes(x) ::: x jokes about being addicted to caffeine
> Unaware(x) ::: x is unaware that caffeine is a drug
> Student(x) ::: x is a student
>
> Premises:
> forall x (Drinks(x) implies Dependent(x)) ::: All coffee drinkers are
>     dependent on caffeine
> forall x (Drinks(x) xor Jokes(x)) ::: People either drink coffee or joke
>     about caffeine addiction
> forall x (Jokes(x) implies not Unaware(x)) ::: Those who joke are aware
>     caffeine is a drug
> (Student(rina) and Unaware(rina)) xor not (Student(rina) or Unaware(rina))
>     ::: Rina's student and awareness status
>
> Conclusion:
> Jokes(rina) xor Unaware(rina) ::: Rina jokes about caffeine or is unaware
>     it is a drug
> ```

# G. Few-Shot Examples for Constrained Few-Shot Baseline

This section presents the in-context examples used in the Constrained Few-Shot baseline for each dataset. The below examples are used to instruct the LLM about the required structure rather than the solution guidance.

## G.1. GSM8K Few-Shot Examples

**Example 1**
```
{
  "steps": [
    "Natalia sold 48 clips in April.",
    "She sold half as many clips in May, which is 24.",
    "Adding April and May sales gives 48 + 24."
  ],
  "answer": "72"
}
```
---
**Example 2**
```
{
  "steps": [
    "The book has 120 pages total.",
    "Julie read 12 pages yesterday and 24 pages today.",
    "She has read 36 pages, leaving 84 pages.",
    "Half of the remaining pages is 42."
  ],
  "answer": "42"
}
```

## G.2. MATH500 Few-Shot Examples

**Example 1**
```
{
  "steps": [
    "Continuity at $x=2$ requires $a(2)+3 = 2-5$.",
    "Solving gives $2a+3=-3$ so $a=-3$.",
    "Continuity at $x=-2$ requires $2(-2)-b = -2-5$.",
    "Solving gives $-4-b=-7$ so $b=3$.",
    "Thus $a+b=-3+3$."
  ],
  "answer": "0"
}
```
---
**Example 2**
```
{
  "steps": [
    "Let $w$ be the number of days Sam worked.",
    "He did not work $20-w$ days.",
    "Total earnings are $60w-30(20-w)=660$.",
    "Solving gives $90w=1260$ so $w=14$.",
    "Days not worked are $20-14$."
  ],
  "answer": "6"
}
```

## G.3. GSM-Symbolic Few-Shot Examples

```
<<tf - t>>
<<c + nc>>
<<ch1 + ch2 - a>>
<<l1 - g>>
<<t + tm + td>>
<<c + nc * (d2 - d1 + 1)>>
<<gb1 - l1 - l2>>
<<m - q * p>>
```

## G.4. FOLIO Few-Shot Examples

**Example 1: Coffee and Caffeine**

```
Predicates:
Dependent(x) ::: x is a person dependent on caffeine.
Drinks(x) ::: x regularly drinks coffee.
Jokes(x) ::: x jokes about being addicted to caffeine.
Unaware(x) ::: x is unaware that caffeine is a drug.
Student(x) ::: x is a student.

Premises:
forall x (Drinks(x) implies Dependent(x)) ::: All people who regularly
  drink coffee are dependent on caffeine.
forall x (Drinks(x) xor Jokes(x)) ::: People either regularly drink
  coffee or joke about being addicted to caffeine.
forall x (Jokes(x) implies not Unaware(x)) ::: No one who jokes about
  being addicted to caffeine is unaware that caffeine is a drug.
(Student(rina) and Unaware(rina)) xor not (Student(rina) or Unaware(rina))
  ::: Rina is either a student and unaware that caffeine is a drug, or
  neither a student nor unaware that caffeine is a drug.

Conclusion:
Jokes(rina) xor Unaware(rina) ::: Rina is either a person who jokes about
  being addicted to caffeine or is unaware that caffeine is a drug.
```

**Example 2: Even and Odd Integers**

```
Predicates:
Even(x) ::: x is even.
Odd(x) ::: x is odd.
Integer(x) ::: x is an integer.
Sum(x, y, z) ::: z is the sum of x and y.

Premises:
Integer(a) and Integer(b) ::: a and b are integers.
Even(a) ::: a is even.
Odd(b) ::: b is odd.
forall x (forall y ((Even(x) and Odd(y)) implies Odd(x+y))) ::: The sum
  of an even integer and an odd integer is odd.

Conclusion:
Odd(a+b) ::: a+b is odd.
```

---

Example 2

**Prompt**

John drives for 3 hours at a speed of 60 mph and then turns around because he realizes he forgot something very important at home. He tries to get home in 4 hours but spends the first 2 hours in standstill traffic. He spends the next half-hour driving at a speed of 30 mph, before being able to drive the remaining time of the 4 hours going at 80 mph. How far is he from home at the end of those 4 hours?

---

**Solution**

**Gold:**

$$3 \cdot 60 = 180 \text{ miles}$$

So when he turns around, he is 180 miles from home.
He is stuck for 2 hours, so he can only drive for:

$$4 - 2 = 2 \text{ hours.}$$

In the next half-hour at 30 mph, he travels:

$$30 \cdot 0.5 = 15 \text{ miles.}$$

He then has:

$$2 - 0.5 = 1.5 \text{ hours}$$

left, and at 80 mph he travels:

$$80 \cdot 1.5 = 120 \text{ miles.}$$

So total return distance is:

$$15 + 120 = 135 \text{ miles.}$$

Thus, distance still remaining from home is:

$$180 - 135 = 45.$$

**Answer:** 45

**Prediction:**

```
{
    "steps": [
        "John drives for 3 hours at 60 mph, so he covers 60 * 3 = 180 miles.",
        "He turns around and tries to get home in 4 hours,
         but spends the first 2 hours stuck in traffic.",
        "He then drives for half an hour at 30 mph,
         so he covers 30 * 0.5 = 15 miles.",
        "In the last 2.5 hours, he drives at 80 mph,
         so he covers 80 * 2.5 = 200 miles.",
        "Total distance covered is 180 + 15 + 200 = 395 miles.",
        "Since he turns around, he is 395 miles away from home."
    ],
    "answer": "395"
}
```

**Correct:** false

---

### Example 3

**Prompt**

At {name}'s house, there are {mult} as many {food1}s as {food2}s. He has a total of {n} {food2}s in his house. {name} bought {m} more {food2}s at the store and {k} fewer {food1}s than the number of {food2}s. Find the combined total of the number of {food1}s and {food2}s {name} has in the house.

---

**Solution**
**Gold:**

$$n \cdot mult + n + 2m - k$$

**Prediction:**

```
m + n + k + n
```

**Correct:** false

---

### Example 4

**Prompt**

There are {n} girls in the {place}. If there are {mult} times the number of boys in the {place}, how many kids are in the {place}?

---

**Solution**
**Gold:**

$$n \cdot (mult + 1)$$

**Prediction:**

```
n + mult * n
```

**Correct:** true

Example 5

**Prompt**
**Problem:**
When the Monkeypox virus occurs in a being, it may get Monkeypox.
Monkeypox virus can occur in certain animals.
Humans are mammals.
Mammals are animals.
Symptoms of Monkeypox include fever, headache, muscle pains, and tiredness.
People feel tired when they get the flu.

**Question:**
Based on the above information, is the following statement true, false, or uncertain?

*There is an animal.*

---

**Solution**
**Gold:** True
**Raw Prediction:**

```
First, let's parse the problem and question into first-order logic formulas.

Predicates:
- Monkeypox(x): x has Monkeypox.
- OccursIn(x, y): the Monkeypox virus occurs in y.
- Animal(x): x is an animal.
- Mammal(x): x is a mammal.
- Human(x): x is a human.
- Fever(x): x has a fever.
- Headache(x): x has a headache.
- MusclePains(x): x has muscle pains.
- Tiredness(x): x is tired.
- Flu(x): x has the flu.

Premises:
- There exists an animal because Monkeypox virus can occur in certain animals.

Question:
There is an animal.

Conclusion:
True
```

**Prediction:** null

# H. System Prompts for Constrained Prompting Baseline

This section presents the carefully engineered system prompts used in the Constrained Prompting baseline for each dataset. These prompts specify the required output structure and format constraints.

## H.1. GSM8K and MATH500 System Prompt

```
You are a meticulous math tutor. Solve the problem step by step and
OUTPUT ONLY a single JSON object conforming to the provided schema,
'{steps: string[], answer: string}'. No extra text, no code fences,
no commentary outside JSON.

{
  "steps": [
    "Step 1",
    "Step 2",
    "...",
    "Step N"
  ],
  "answer": "Final Answer"
}

Result in 'answer'. No extra text, no code fences, no commentary
outside JSON.
```

## H.2. GSM-Symbolic System Prompt

```
Only output the symbolic expression wrapped in [[START]] [[END]] that
answers the question. The expression must use numbers as well as the
variables defined in the question. You are only allowed to use the
following operations: +, -, /, //, %, (), and int().

You will always respond in the format described below:
[[START]]symbolic expression[[END]]
```

## H.3. FOLIO System Prompt

```
Given a problem description and a question. The task is to parse the
problem and the question into first-order logic formulas.

The grammar of the first-order logic formula is defined as follows:
1) logical conjunction of expr1 and expr2: expr1 {and} expr2
2) logical disjunction of expr1 and expr2: expr1 {or} expr2
3) logical exclusive disjunction of expr1 and expr2: expr1 {xor} expr2
4) logical negation of expr1: {not}expr1
5) expr1 implies expr2: expr1 {implies} expr2
6) expr1 if and only if expr2: expr1 {iff} expr2
7) logical universal quantification: {forall} x
8) logical existential quantification: {exists} x

These are the ONLY operations in the grammar.

Expected output format:
Predicates:
P(x) ::: description of P
Q(x) ::: description of Q

Premises:
forall x (P(x) implies Q(x)) ::: premise description

Conclusion:
Q(a) ::: conclusion statement

------
Given the message above, Just write the "Predicates:", "Premises:",
and "Conclusion:" fields. DO NOT OUTPUT ANYTHING ELSE OTHER THAN THE
ANSWER! Answer the question EXACTLY like the examples.
```

# I. Constraints for Constrained Decoding Baseline

This section presents the formal constraints used in the Constrained Decoding baseline with XGrammar. For JSON-based tasks, we use Pydantic schemas, while symbolic and logical tasks use context-free grammars.

### I.1. GSM8K Schema

```
class GSM8KSchema(BaseModel):
    steps: Annotated[List[str], Field(min_length=1)]
    answer: Annotated[str, Field(min_length=1)]
```

### I.2. MATH500 Schema

```
class MATHSchema(BaseModel):
    steps: Annotated[List[str], Field(min_length=1)]
    answer: Annotated[str, Field(min_length=1)]
```

### I.3. GSM-Symbolic Grammar

```
start: space? "<" "<" space? expr space? ">" ">" space?

expr: expr space? "+" space? term
    | expr space? "-" space? term
    | term

term: term space? "*" space? factor
    | term space? "/" space? factor
    | term space? "//" space? factor
    | term space? "%" space? factor
    | factor space?

factor: "-" space? factor
      | TYPE "(" space? expr space? ")"
      | primary space?

primary: NUMBER
       | VARIABLE
       | "(" space? expr space? ")"

TYPE: "int"

space: " "

%import common.CNAME -> VARIABLE
%import common.NUMBER
```

### I.4. FOLIO Grammar

```
start: predicate_section premise_section conclusion_section

predicate_section: "Predicates:" predicate_definition+
premise_section: "Premises:" premise+
conclusion_section: "Conclusion:" conclusion+

predicate_definition: PREDICATE "(" VAR ("," VAR)* ")" COMMENT
                      -> define_predicate
premise: quantified_expr COMMENT -> define_premise
conclusion: quantified_expr COMMENT -> define_conclusion

quantified_expr: quantifier VAR "(" expression ")" | expression
quantifier: "∀" -> forall | "∃" -> exists

expression: bimplication_expr

?bimplication_expr: implication_expr ("↔" bimplication_expr)?  -> iff
?implication_expr: xor_expr ("→" implication_expr)?  -> imply
?xor_expr: or_expr ("⊕" xor_expr)?  -> xor
?or_expr: and_expr ("∨" or_expr)?  -> or
?and_expr: neg_expr ("∧" and_expr)?  -> and
?neg_expr: "¬" quantified_expr  -> neg
        | atom

?atom: PREDICATE "(" VAR ("," VAR)* ")" -> predicate
    | "(" quantified_expr ")"

// Variable names begin with a lowercase letter
VAR: /[a-z][a-zA-Z0-9_]*/  | /[0-9]+/

// Predicate names begin with a capital letter
PREDICATE: /[A-Z][a-zA-Z0-9]*/

COMMENT: /:::.*\n/

%import common.WS
%ignore WS
```

---

**Example 1**

**Prompt**

If $f(x) = \dfrac{3x-2}{x-2}$, what is the value of $f(-2) + f(-1) + f(0)$? Express your answer as a common fraction.

---

**Solution**
**Gold:**

$$f(-2) + f(-1) + f(0) = \frac{3(-2)-2}{-2-2} + \frac{3(-1)-2}{-1-2} + \frac{3(0)-2}{0-2} = \frac{-8}{-4} + \frac{-5}{-3} + \frac{-2}{-2} = 2 + \frac{5}{3} + 1 = \frac{14}{3}.$$

**Prediction:**

```
{
    "steps": [
        "Calculate f(-2) by substituting -2 into the function:
         f(-2) = (3(-2)-2)/((-2)-2) = (-6-2)/(-4) = (-8)/(-4) = 2.",
        "Calculate f(-1) by substituting -1 into the function:
         f(-1) = (3(-1)-2)/((-1)-2) = (-3-2)/(-3) = (-5)/(-3) = 5/3.",
        "Calculate f(0) by substituting 0 into the function:
         f(0) = (3(0)-2)/(0-2) = (-2)/(-2) = 1.",
        "Add them:
         2 + 5/3 + 1 = 14/3."
    ],
    "answer": "14/3"
}
```

**Correct:** true

# J. Case Study: Why Draft-Conditioned Constrained Decoding Works

This section presents a detailed example demonstrating why Draft-Conditioned Constrained Decoding outperforms other approaches. We show how constrained decoding can harm reasoning quality, while our DCCD approach preserves both correctness and structure.

## J.1. Problem Statement

MATH500 Problem

Define

$$p = \sum_{k=1}^{\infty} \frac{1}{k^2} \quad \text{and} \quad q = \sum_{k=1}^{\infty} \frac{1}{k^3}.$$

Find a way to write

$$\sum_{j=1}^{\infty} \sum_{k=1}^{\infty} \frac{1}{(j+k)^3}$$

in terms of $p$ and $q$.

**Ground Truth Answer:** $\boxed{p - q}$

## J.2. Baseline Predictions

### J.2.1. CONSTRAINED DECODING (XGRAMMAR) - INCORRECT

```
{
  "steps": [
    "Given the sums p and q, we start by examining the double sum ...",
    "We can change the order of summation. Let's denote m = j + k. For each m, j can
     range from 1 to m - 1.",
    "Thus, we can rewrite the double sum as ...",
    "Now, let's expand the inner sum. We need to find ...",
    "The inner sum is a constant 1/m3 summed m-1 times, thus it equals (m-1) / m3.",
    "So, the expression becomes ...",
    "We can split the fraction ...",
    "This can be written as ...",
    "We know that the sum of 1/m3 is q and the sum of 1/m4 is a term related to the
    Riemann zeta function (4) = p^2/3.",
    "Therefore, the expression becomes q - p^2/3.",
    "The final answer is q - p^2/3."
  ],
  "answer": "q - p^2/3"
}
```

**Analysis:** The model produces valid JSON but makes a critical mathematical error: it incorrectly claims that $\sum_{m=2}^{\infty} \frac{1}{m^4} = \frac{p^2}{3}$, leading to the wrong answer $q - \frac{p^2}{3}$.

### J.2.2. CONSTRAINED PROMPTING - INCOMPLETE/MALFORMED

```
{
  "steps": [
    "Express the double sum as a sum over all pairs (j, k) where j and k are
    positive integers.",
    "Notice that each term 1/(j+k)^3 appears in the sum when j and k are such that j
     + k = n, for n ranging from 2 to infinity.",
    "For each n, the term 1/n^3 appears in the sum for all pairs (j, k) such that j
    + k = n, specifically for j = 1 to n-1.",
    "Thus, we can write the double sum as a sum over n from 2 to infinity, where the
     contribution for each n is 1/n^3 times the number of pairs (j, k) such that j +
     k = n.",
    "The number of such pairs is n-1, giving us sum_{n=2}^infty (n-1)/n^3.",
    "Rewrite the sum as sum_{n=2}^infty (n-1)/n^3 = sum_{n=2}^infty (1/n^2 - 1/n^3)",

    "Separate the sums: sum_{n=2}^infty 1/n^2 - sum_{n=2}^infty 1/n^3.",
    "Recognize that sum_{n=2}^infty 1/n^2 = p - 1 and sum_{n=2}^infty 1/n^3 = q -
    1.",
    "Therefore, the final expression is (p - 1) - (q
```

**Analysis:** The reasoning is mathematically correct and would lead to the right answer $(p - 1) - (q - 1) = p - q$, but the output is truncated mid-sentence and produces invalid JSON. The structure constraint is violated.

### J.2.3. CONSTRAINED FEW-SHOT - INCOMPLETE/MALFORMED

```
{
  "steps": [
    "First, consider the double sum $\sum_{j=1}^\infty \sum_{k=1}^\infty \frac{1}{(j
    +k)^3}$. We can change the order of summation to make the expression more
    manageable.",
    "Let $n = j + k$. Then, for a fixed $n$, $j$ ranges from $1$ to $n-1$ and $k = n
     - j$. Therefore, the double sum can be rewritten as $\sum_{n=2}^\infty \sum_{j
    =1}^{n-1} \frac{1}{n^3}$.",
    "The inner sum is $\sum_{j=1}^{n-1} \frac{1}{n^3} = \frac{n-1}{n^3} = \frac{1}{n
    ^2} - \frac{1}{n^3}$. Thus the double sum becomes $\sum_{n=2}^\infty \left( \
    frac{1}{n^2} - \frac{1}{n^3} \right)$.",
    "This can be split into two separate sums: $\sum_{n=2}^\infty \frac{1}{n^2} - \
    sum_{n=2}^\infty \frac{1}{n^3}$.",
    "Recognize that $\sum_{n=2}^\infty \frac{1}{n^2} = \sum_{n=1}^\infty \frac{1}{n
    ^2} - \frac{1}{1^2} = p - 1$ and $\sum_{n=2}^\infty \frac{1}{n^3} = \sum_{n=1}^\
    infty \frac{1}{n^3} - \frac{1}{
```

**Analysis:** Despite providing few-shot examples, the model output is truncated mid-expression within the last step. The reasoning up to the truncation point is mathematically correct and would lead to the answer $(p-1) - (q-1) = p - q$, but the output produces invalid JSON due to incomplete string and missing closing brackets. Both the answer field and proper JSON structure are absent.

## J.3. Draft-Conditioned Constrained Decoding

### J.3.1. STAGE 1: UNCONSTRAINED REASONING GENERATION

> **Stage 1 Output**
>
> To find a way to express $\sum_{j=1}^{\infty} \sum_{k=1}^{\infty} \frac{1}{(j+k)^3}$ in terms of $p$ and $q$, we need to carefully analyze and manipulate the given double sum. Let's start by examining the structure of the sum.
>
> First, let's rewrite the double sum in a more convenient form:
>
> $$\sum_{j=1}^{\infty} \sum_{k=1}^{\infty} \frac{1}{(j+k)^3}.$$
>
> We can change the order of summation. To do this, we need to consider the range of $j + k$. For a fixed $n = j + k$, $j$ can range from 1 to $n - 1$ and $k$ can range from 1 to $n - 1$. Therefore, we can rewrite the sum as:
>
> $$\sum_{n=2}^{\infty} \sum_{j=1}^{n-1} \frac{1}{n^3}.$$
>
> Now, let's simplify the inner sum:
>
> $$\sum_{j=1}^{n-1} \frac{1}{n^3} = \frac{1}{n^3} \sum_{j=1}^{n-1} 1 = \frac{1}{n^3}(n-1) = \frac{n-1}{n^3}.$$
>
> Substituting this back into the outer sum, we get:
>
> $$\sum_{n=2}^{\infty} \frac{n-1}{n^3}.$$
>
> We can split the fraction:
>
> $$\sum_{n=2}^{\infty} \frac{n-1}{n^3} = \sum_{n=2}^{\infty} \left( \frac{n}{n^3} - \frac{1}{n^3} \right) = \sum_{n=2}^{\infty} \frac{1}{n^2} - \sum_{n=2}^{\infty} \frac{1}{n^3}.$$
>
> We can separate the sums:
>
> $$\sum_{n=2}^{\infty} \frac{1}{n^2} = \left( \sum_{n=1}^{\infty} \frac{1}{n^2} \right) - 1 = p - 1,$$
>
> $$\sum_{n=2}^{\infty} \frac{1}{n^3} = \left( \sum_{n=1}^{\infty} \frac{1}{n^3} \right) - 1 = q - 1.$$
>
> Substituting these results back into our expression, we get:
>
> $$\sum_{n=2}^{\infty} \frac{n-1}{n^3} = (p-1) - (q-1) = p - q.$$
>
> Thus, the final answer is:
>
> $$\boxed{p - q}.$$

### J.3.2. STAGE 2: STRUCTURE CONVERSION

---

**Stage 2 Output**

```
{
  "steps": [
    "Rewrite the double sum $\sum_{j=1}^\infty \sum_{k=1}^\infty \frac{1}{(j+k)^3}$
    in a more convenient form",
    "Change the order of summation: for fixed $n = j + k$, rewrite as $\sum_{n=2}^\
    infty \sum_{j=1}^{n-1} \frac{1}{n^3}$",
    "Simplify the inner sum: $\sum_{j=1}^{n-1} \frac{1}{n^3} = \frac{n-1}{n^3}$",
    "Substitute back into the outer sum to get $\sum_{n=2}^\infty \frac{n-1}{n^3}$",
    "Split the fraction: $\sum_{n=2}^\infty \frac{n-1}{n^3} = \sum_{n=2}^\infty \
    frac{1}{n^2} - \sum_{n=2}^\infty \frac{1}{n^3}$",
    "Separate the sums: $\sum_{n=2}^\infty \frac{1}{n^2} = p - 1$ and $\sum_{n=2}^\
    infty \frac{1}{n^3} = q - 1$",
    "Substitute the results: $(p - 1) - (q - 1) = p - q$"
  ],
  "answer": "p - q"
}
```

- - - - - - - - - - - - - - - - - - - - - - - - - - - - - - - - - - - - - - - - - - - - -

**Analysis:** Stage 2 successfully converts the unconstrained reasoning from Stage 1 into perfectly valid JSON structure. The model extracts the key reasoning steps, preserves mathematical notation, and produces both the correct answer $p - q$ (CORRECT) and valid JSON structure (VALID). This demonstrates how Draft-Conditioned Constrained Decoding achieves both goals: high-quality reasoning from Stage 1 and strict format compliance from Stage 2.

## J.4. Constrained Few Shot Results

---

**Example 2**

**Prompt**

John drives for 3 hours at a speed of 60 mph and then turns around because he realizes he forgot something very important at home. He tries to get home in 4 hours but spends the first 2 hours in standstill traffic. He spends the next half-hour driving at a speed of 30 mph, before being able to drive the remaining time of the 4 hours going at 80 mph. How far is he from home at the end of those 4 hours?

---

**Solution**
**Gold:**

$$3 \cdot 60 = 180 \text{ miles}$$

So when he turns around, he is 180 miles from home.
He is stuck for 2 hours, so he can only drive for:

$$4 - 2 = 2 \text{ hours.}$$

In the next half-hour at 30 mph, he travels:

$$30 \cdot 0.5 = 15 \text{ miles.}$$

He then has:
$$2 - 0.5 = 1.5 \text{ hours}$$

left, and at 80 mph he travels:
$$80 \cdot 1.5 = 120 \text{ miles.}$$

So total return distance is:
$$15 + 120 = 135 \text{ miles.}$$

Thus, distance still remaining from home is:

$$180 - 135 = 45.$$

**Answer:** 45

**Prediction:**

```
{
    "steps": [
        "John drives for 3 hours at 60 mph, so he covers 60 * 3 = 180 miles.",
        "He turns around and tries to get home in 4 hours,
         but spends the first 2 hours stuck in traffic.",
        "He then drives for half an hour at 30 mph,
         so he covers 30 * 0.5 = 15 miles.",
        "In the last 2.5 hours, he drives at 80 mph,
         so he covers 80 * 2.5 = 200 miles.",
        "Total distance covered is 180 + 15 + 200 = 395 miles.",
        "Since he turns around, he is 395 miles away from home."
    ],
    "answer": "395"
}
```

**Correct:** false

---

### Example 3

**Prompt**

At {name}'s house, there are {mult} as many {food1}s as {food2}s. He has a total of {n} {food2}s in his house. {name} bought {m} more {food2}s at the store and {k} fewer {food1}s than the number of {food2}s. Find the combined total of the number of {food1}s and {food2}s {name} has in the house.

---

**Solution**
**Gold:**

$$n \cdot mult + n + 2m - k$$

**Prediction:**

```
m + n + k + n
```

**Correct:** false

---

### Example 4

**Prompt**

There are {n} girls in the {place}. If there are {mult} times the number of boys in the {place}, how many kids are in the {place}?

---

**Solution**
**Gold:**

$$n \cdot (mult + 1)$$

**Prediction:**

```
n + mult * n
```

**Correct:** true

---

Example 5

---

**Prompt**
**Problem:**
When the Monkeypox virus occurs in a being, it may get Monkeypox.
Monkeypox virus can occur in certain animals.
Humans are mammals.
Mammals are animals.
Symptoms of Monkeypox include fever, headache, muscle pains, and tiredness.
People feel tired when they get the flu.

**Question:**
Based on the above information, is the following statement true, false, or uncertain?

*There is an animal.*

---

**Solution**
**Gold:** True
**Raw Prediction:**

```
First, let's parse the problem and question into first-order logic formulas.

Predicates:
- Monkeypox(x): x has Monkeypox.
- OccursIn(x, y): the Monkeypox virus occurs in y.
- Animal(x): x is an animal.
- Mammal(x): x is a mammal.
- Human(x): x is a human.
- Fever(x): x has a fever.
- Headache(x): x has a headache.
- MusclePains(x): x has muscle pains.
- Tiredness(x): x is tired.
- Flu(x): x has the flu.

Premises:
- There exists an animal because Monkeypox virus can occur in certain animals.

Question:
There is an animal.

Conclusion:
True
```

**Prediction:** null

## K. Additional Case Study: GSM-Symbolic

---

**GSM-Symbolic Problem**

{name} was preparing for a {event} at her house, where she intended to serve {food}. She noticed that she was out of plastic {obj}, so she bought a new package of {obj}. Later, her husband also bought a package of {n1} new {obj} and gave them to {name}. While {name} was making the {food}, she used {n2} of the {obj} to sample her {food}. Later, when she went to set the table, she had a total of {total} {obj}. How many {obj} were in the package that {name} bought?

**Ground Truth:** `total + n2 - n1`

> **Baseline Predictions**
>
> **Constrained Decoding (XGrammar): INCORRECT**
> $\ll total - n2 + n1 \gg$
>
> ---
>
> **Constrained Prompting: INCORRECT**
> $\ll total - n2 - n1 \gg$
>
> ---
>
> **Constrained Few-Shot: INCORRECT**
> $\ll total - n1 + n2 \gg$

---

**Draft-Conditioned Constrained Decoding: Stage 1 - Unconstrained Reasoning**

Let's denote the number of {obj} that {name} bought as $x$.
Initially, {name} bought $x$ {obj}.
Then her husband bought and gave her {n1} more {obj}, so she now has:

$$x + \texttt{n1}$$

While making the {food}, she used {n2} of the {obj}, so the number remaining is:

$$x + \texttt{n1} - \texttt{n2}$$

We're told that when she went to set the table, she had a total of {total} {obj}. Therefore:

$$x + \texttt{n1} - \texttt{n2} = \texttt{total}$$

Solving for $x$:

$$x = \texttt{total} - \texttt{n1} + \texttt{n2}$$

Rearranging to match standard form:

$$x = \texttt{total} + \texttt{n2} - \texttt{n1}$$

Thus, the answer is: `total + n2 - n1`

- - - - - - - - - - - - - - - - - - - - - - - - - - - - - - - - -

**Stage 2 - Structure Conversion: CORRECT**

`<<total + n2 - n1>>`

---

## L. Additional Analysis

### L.1. Efficiency: Accuracy–Adherence–Latency Tradeoff

DCCD requires two end-to-end inference passes per query, which increases wall-clock latency compared to single-pass decoding. We argue that, in structured generation, latency should be interpreted *jointly* with output validity and task accuracy:

a fast method that fails to satisfy the required schema is not directly usable downstream. The relevant comparison is therefore the overall tradeoff among accuracy, adherence, and end-to-end runtime.

Table 5 reports this tradeoff on MATH500, measured with the vLLM inference engine in batched mode on a single NVIDIA H100. Across all four model scales, DCCD incurs only a modest wall-clock overhead over direct CD while delivering substantially higher structured accuracy and maintaining near-perfect adherence. For example, on Qwen2.5-7B, DCCD adds roughly 0.2 minutes over CD on the full benchmark (an overhead of $\sim$13%) while improving accuracy from 37.8 to 53.6 and reaching 100% adherence. The unconstrained baseline is fastest but has zero adherence by construction and is therefore not directly comparable for downstream structured use.

*Table 5.* Accuracy, adherence, and total wall-clock time on MATH500 (vLLM, batched inference, single NVIDIA H100). DCCD is slower than direct CD by a modest margin, but consistently achieves much higher structured accuracy while maintaining near-perfect adherence.

| Model | Method | Accuracy ↑ | Adherence ↑ | Time (min) ↓ |
|---|---|---|---|---|
| Llama-3.2-1B | Unconstrained | 16.40 | 0.00 | **0.294** |
| | Constrained | 5.20 | 90.80 | 0.999 |
| | DCCD (ours) | **13.80** | **96.80** | 1.358 |
| Qwen2.5-1.5B | Unconstrained | 40.00 | 0.00 | **0.376** |
| | Constrained | 14.80 | 96.40 | 0.892 |
| | DCCD (ours) | **34.60** | **98.60** | 1.298 |
| Qwen2.5-3B | Unconstrained | 51.60 | 0.00 | **0.205** |
| | Constrained | 31.40 | 96.20 | 1.190 |
| | DCCD (ours) | **44.20** | **98.40** | 1.445 |
| Qwen2.5-7B | Unconstrained | 68.00 | 0.00 | **0.148** |
| | Constrained | 37.80 | 98.00 | 1.573 |
| | DCCD (ours) | **53.60** | **100.00** | 1.781 |

In latency-sensitive settings where strict schema adherence is required, DCCD therefore provides a favorable operating point on the accuracy–latency frontier: it nearly closes the gap to unconstrained accuracy while paying only a small multiplicative cost over single-pass constrained decoding.

### L.2. Draft Quality, Draft Length, and the Effect on Final Output

Because the second stage of DCCD conditions on the stage-1 draft, the semantic quality of that draft directly shapes the final structured output. We examine three related questions: (i) does draft quality bound the final accuracy of DCCD? (ii) is DCCD's gain explained by giving it more total tokens to generate? and (iii) how short can the draft be before DCCD loses its advantage?

**Draft quality as an upper bound.** Table 6 evaluates the unconstrained stage-1 draft as a diagnostic upper bound on what DCCD could in principle realize under the structural constraint. Two consistent patterns emerge. First, the unconstrained draft accuracy acts as an effective ceiling on DCCD: when the draft is stronger (e.g., 76.0 for Qwen2.5-14B), DCCD reaches a correspondingly higher final accuracy (58.6), while weaker drafts (e.g., 24.6 for Llama-3.2-1B) bound DCCD to a lower value (19.8). This confirms that draft errors propagate to the constrained output. Second, DCCD recovers a substantial fraction of the accuracy lost by directly applying CD. On the 14B model, CD drops from a 76.0 draft to 47.6 under constraints (a loss of 28.4 points); DCCD recovers 11.0 of those points, reaching 58.6. Similar recovery patterns hold across all six models we evaluated.

**Budget-matched comparison.** A reasonable concern is that DCCD's gain might be explained simply by generating more tokens overall. To control for this, we compare CD and DCCD under the *same* total generation budget, counting both stage 1 and stage 2 tokens for DCCD. As shown in Table 7, DCCD remains substantially more accurate than CD even when the total budget is held fixed (with DCCD splitting the budget equally between stage 1 and stage 2).

**Draft-length ablation.** Table 8 caps the stage-1 draft at fixed token lengths and reports DCCD accuracy on the 8B model. DCCD retains an advantage over CD even with a short 256-token draft, and accuracy increases monotonically with draft length, plateauing around 1024–2048 tokens.

*Table 6.* MATH500 accuracy across model scales. The unconstrained draft column characterizes the semantic quality of stage 1; it does not satisfy the schema and is not a usable structured output. DCCD consistently recovers a large fraction of the accuracy gap that direct CD loses relative to the draft.

| Model | Unconstrained Draft ↑ | CD ↑ | DCCD (ours) ↑ |
|---|---|---|---|
| Llama-3.2-1B Instruct | 24.6 | 6.0 | **19.8** |
| Qwen2.5-1.5B Instruct | 45.6 | 15.0 | **38.2** |
| Qwen2.5-3B Instruct | 54.2 | 33.4 | **46.8** |
| Qwen2.5-7B Instruct | 71.4 | 43.6 | **52.8** |
| Llama-3.1-8B Instruct | 39.2 | 28.6 | **35.0** |
| Qwen2.5-14B Instruct | 76.0 | 47.6 | **58.6** |

*Table 7.* Budget-matched comparison between CD and DCCD on MATH500. DCCD's total budget is split equally between stage 1 and stage 2.

| Model | Budget | CD | DCCD |
|---|---|---|---|
| 1B | 900 | 6.40 | **15.80** |
| 1.5B | 600 | 14.00 | **17.40** |
| 8B | 900 | 28.60 | **32.20** |

*Table 8.* DCCD draft-length ablation on MATH500 using the 8B model. DCCD retains an advantage over CD even with very short drafts.

| Model | CD | 256 | 512 | 1024 | 2048 |
|---|---|---|---|---|---|
| 8B | 28.60 | 29.20 | 30.50 | 34.80 | **35.00** |

Taken together, these three analyses support two conclusions: (i) DCCD is fundamentally bounded by the semantic quality of the stage-1 draft, so improvements to the draft model translate directly into improvements in structured output; and (ii) DCCD's gain is not explained by additional generation budget alone, and persists across draft lengths down to short drafts.

**Does Stage 2 preserve or recover correctness?** We further decompose DCCD's behavior by separating cases where the stage-1 draft is already correct from cases where it is incorrect, and asking how often stage 2 reaches the correct final answer in each case (Table 9). Stage 2 preserves a correct draft in the large majority of cases (between 72.9% and 83.6%), and rarely recovers from an incorrect draft (between 0.2% and 3.3%). This is consistent with the view that DCCD's role is structured *realization* of the draft's semantics rather than autonomous correction.

*Table 9.* Relationship between stage-1 draft correctness and final DCCD output correctness.

| Model | Draft Correct → Final Correct (%) | Draft Wrong → Final Correct (%) |
|---|---|---|
| Llama-3.2-1B Instruct | 72.9 | 0.2 |
| Qwen2.5-1.5B Instruct | 78.5 | 3.3 |
| Qwen2.5-3B Instruct | 83.6 | 2.4 |
| Qwen2.5-7B Instruct | 75.7 | 1.3 |

### L.3. Feasible Mass and Structured Correctness

The cumulative log feasible mass used in Algorithm 1 is *not* a direct measure of reasoning correctness; rather, it measures how much constraint-induced distortion stage 2 incurs when converting a draft into a valid structured output. Our intended claim is that higher cumulative feasible mass corresponds to smaller projection tax in stage 2, which makes the structured realization less distortive and therefore more likely to preserve the semantic content of the draft.

To make this empirically concrete, Table 10 reports both accuracy and feasible mass for CD and DCCD across six models on MATH500. DCCD systematically increases feasible mass over CD, and across models, larger feasible-mass improvements coincide with larger accuracy improvements. The Pearson correlation between cumulative log feasible mass and final structured correctness across sampled drafts is 0.71, indicating a positive and statistically significant association.

We emphasize that feasible-mass scoring is only one practical instantiation of the broader DCCD framework: as noted in the main paper (Line 256, right column), alternative late-selection criteria such as verifier scores, judges, or majority voting can also be plugged into the same two-stage structure.

*Table 10.* Feasible mass and structured accuracy of CD and DCCD on MATH500 across model scales.

| Model | Acc. (CD) | Acc. (DCCD) | Feasible Mass (CD) | Feasible Mass (DCCD) |
|---|---|---|---|---|
| Llama-3.2-1B Instruct | 0.0660 | 0.1260 | 0.4563 | 0.8181 |
| Qwen2.5-1.5B Instruct | 0.1300 | 0.3260 | 0.6690 | 0.7821 |
| Qwen2.5-3B Instruct | 0.3160 | 0.4300 | 0.6953 | 0.7425 |
| Qwen2.5-7B Instruct | 0.4140 | 0.5280 | 0.8019 | 0.9156 |
| Llama-3.1-8B Instruct | 0.2740 | 0.3340 | 0.7144 | 0.8822 |
| Qwen2.5-14B Instruct | 0.4600 | 0.5720 | 0.7901 | 0.8729 |

### L.4. Multi-Draft Selection: Feasible Mass vs. Majority Voting

When multiple stage-1 drafts are sampled, the second stage can use different rules to select which draft to commit to. Table 11 compares feasible-mass selection against majority voting at $K = 5$ drafts on MATH500. Majority voting modestly outperforms feasible-mass selection in both settings tested, which is consistent with the view that feasible mass is a proxy for distortion rather than a direct correctness signal. This positions feasible-mass selection as a lightweight default that does not require a separate verifier or many additional samples, while leaving room to plug in stronger selection rules where the additional compute is acceptable.

*Table 11.* Multi-draft selection rules on MATH500 with $K = 5$ stage-1 drafts.

| Model | Feasible-Mass Selection (%) | Majority Voting (%) |
|---|---|---|
| 3B | 47.0 | **50.0** |
| 8B | 35.2 | **39.4** |

### L.5. Effect of Schema Tightness

The feasible-mass view predicts that tighter output constraints should increase the distortion introduced by direct CD, because they shrink the set of valid continuations at each step. We test this on GSM8K by comparing two schemas: a flexible schema in which the `answer` field is a string, and a stricter schema in which the `answer` field must be an integer (Table 12). Tightening the schema drops CD accuracy in both models, with a substantial 18.58-point drop on the 3B model. This is consistent with the prediction that tighter schemas amplify the projection tax that motivates DCCD.

*Table 12.* Effect of schema tightness on CD accuracy on GSM8K.

| Model | CD (flexible: answer as string) | CD (strict: answer as int) | Drop |
|---|---|---|---|
| Qwen2.5-1.5B Instruct | 49.36 | 45.94 | 3.42 |
| Qwen2.5-3B Instruct | 81.58 | 63.00 | 18.58 |

### L.6. Comparison with Thinking-Token Generation

DCCD shares a high-level intuition with thinking-token approaches: semantic reasoning should happen before the final structured answer is produced. However, DCCD makes this separation *explicit*, generating an unconstrained draft in stage 1 and applying CD only in stage 2. To test whether this explicit separation matters, we compare a reasoning-capable 1.5B model with CD applied only at the answer stage against DCCD using the same base model as the stage-1 drafter (Table 13). Applying CD at the answer stage degrades performance from 63.2 to 51.2, while DCCD largely preserves the unstructured-answer accuracy at 62.6.

This suggests that the gain from DCCD is not merely from giving the model a chance to reason before answering, but specifically from keeping that reasoning *unconstrained*: applying CD even only at the final answer stage already introduces enough projection tax to substantially hurt accuracy.

*Table 13.* Thinking-style generation vs. DCCD on MATH500, using a reasoning-capable 1.5B model.

| Model | Dataset | Unstructured Answer | Thinking + CD at Answer Stage | DCCD (R1 1.5B + Qwen 2.5 1.5B) |
|---|---|---|---|---|
| Deepseek R1 Qwen 1.5B | MATH500 | 63.2 | 51.2 | **62.6** |

*(a)* GSM8K benchmark scaling results

*(b)* MATH500 benchmark scaling results.

*Figure 12.* Detailed test-time scaling comparison across all six models for GSM8K and MATH500 benchmarks. Each subplot shows the scaling behavior for a specific model, comparing Constrained Decoding(CD) (red) with Draft-Conditioned Constrained Decoding(DCCD) (blue). The consistent gap between methods across model sizes and benchmarks demonstrates the robustness of the DCCD approach.

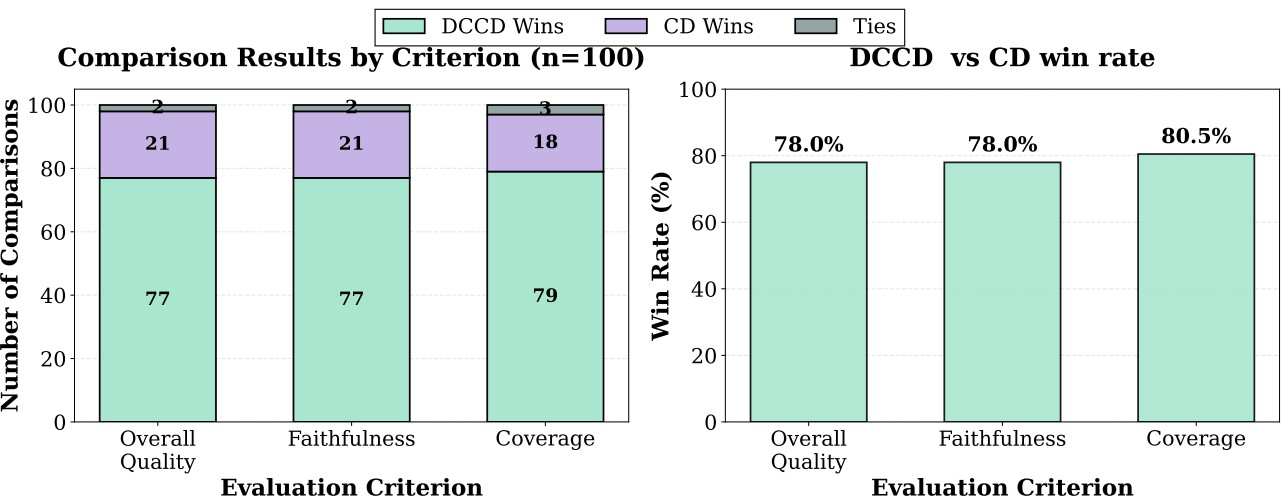

*Figure 13.* DCCD demonstrates consistent superiority across all evaluation criteria with win rates of 78–80.5%. The highest performance on coverage (80.5%) validates our hypothesis that separating reasoning from formatting preserves information density. The low tie rate (2–3%) indicates clear quality differences, while robust performance across diverse prompt categories demonstrates generalizability. These findings confirm that explicit separation of content generation from constraint satisfaction maintains both reasoning quality and information completeness under hard token budgets.

## M. Additional Figures

We show representative figures illustrating how cumulative KL drift grows over the sequence length for both DCCD and standard CD. This KL drift can push the final response toward a lower-likelihood region of the base model distribution. Since the base model is trained to assign higher likelihood to more accurate responses, excessive drift from this distribution can reduce answer correctness. In contrast, DCCD reduces this drift by conditioning decoding on a draft, allowing the final output distribution to remain closer to the base policy.

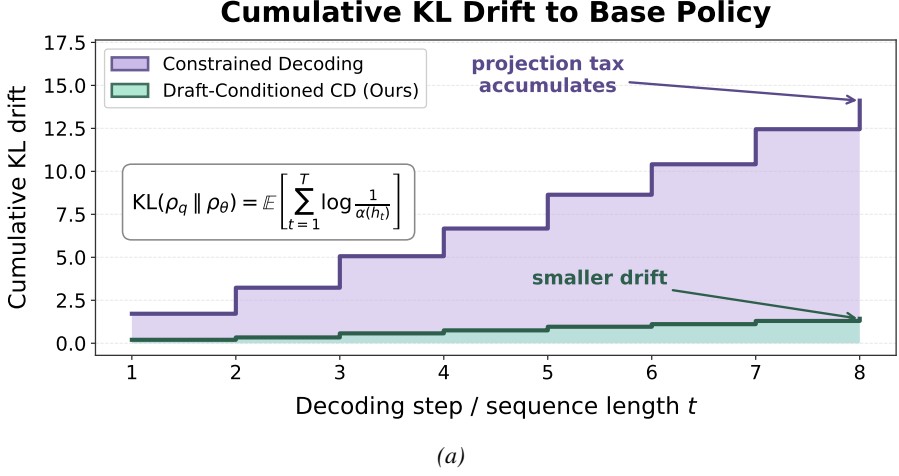

*(a)*

*Distributional Shift from Base Policy*

*(b)*

*Figure 15.* **Low feasible mass can induce distributional distortion.** We illustrate how constrained decoding can accumulate drift from the base policy when the feasible token set receives limited probability mass under the model. **(a)** Repeated renormalization over the feasible set causes a monotonically increasing KL divergence from the base policy across decoding steps, which we refer to as a "projection tax." DCCD reduces this effect by conditioning decoding on a draft, leading to substantially smaller cumulative drift. **(b)** The same phenomenon appears in semantic space: standard CD shifts the final distribution away from the intended target, whereas DCCD remains closer to the base policy and the target region.

