# OpenReview forum: "The Hidden Cost of Structured Generation in LLMs: Draft-Conditioned Constrained Decoding"
_ICML.cc/2026/Conference — ICML 2026 regular_

### Official Review · Reviewer_NV7k · 2026-03-08

**Soundness:** 2
**Presentation:** 3
**Significance:** 1
**Originality:** 1
**Overall Recommendation:** 3
**Confidence:** 4

**Summary:**

Following recent constrainted decoding literature, the paper shows and discusses from KL projection perspective that grammar constrained (CD) decoding lead to predictive distribution that can be very far away from the original distribution and thus impacting the performance.

As a remedy, the author propose a "predict-and-refine" pipeline that first predicts and add constraints only at the second step. The authors show that such method improves the performance against few-shot or simple contrained decoding methods, and the improvement is particularly large for small models. The authors also perform analysis show that indeed, the predictive probability with the employed method is more confident and "predict-and-refine" method generalizes to other domains beyond strict structured prediction.

**Compliance With Llm Reviewing Policy:**

Affirmed.

**Key Questions For Authors:**

The Appendix G shows the few shot examples. However, in such few shots, one expects each example contains both questions and answers while the few-shot examples only contain answers?

**Limitations:**

yes

**Strengths And Weaknesses:**

- Strengths: The paper is easy to follow and with the proposed approach has demonstrated its effectiveness against several competitive baselines including grammar constrained decoding and few shot grammar constrained decoding.
- Weakness: Although the paper highlights KL-projection perspective as a key contribution, I think the contribution is quite weak because first this normalization issues have been shown by previous works and eq(5) holds but is just another way to express the same observations mathematically. On the empirical improvements, the improvements seem significant. However, as I raised in the question section, I am not able to see exactly how the few-shot constrained decoding is performed; I think the question is important because few-shots share the same motivation as the paper by making the structured prediction familiar to the LLM. Finally, the paper proposed "predict-and-refine" method needs to cite significant LLM works as such methology has been proposed in many other domains; very minorly, constrained decoding has been formulated and experimented with a CFG backbone earlier than what paper cited, see https://aclanthology.org/P16-1127.pdf and https://aclanthology.org/P17-1041.pdf.

---

> ### Author Rebuttal · Authors · 2026-03-31
>
> We thank you for your valuable feedback.
>
> > **Question  1:** The Appendix G shows the few shot examples. However, in such few shots, one expects each example contains both questions and answers while the few-shot examples only contain answers?
>
> **Response to Question 1:** Thank you for pointing this out, and we apologize for the confusion. You are correct that standard few-shot prompting should contain input-output pairs. In our setup, the few-shot baseline uses prompt-response demonstrations, but Appendix G only showed the answer-side structured outputs, which made the presentation ambiguous. The corresponding questions are present in the case-study examples (e.g., Appendix J), and we will revise Appendix G in the final version to include the full prompt-response pairs explicitly. For quick reference, here are some [few shot examples](https://drive.google.com/file/d/1nd22neFiKI5ylVG-OGBWJEoJ0WsYig-Y/view?usp=sharing).
>
> ## Response to Weaknesses
>
> > **Weakness  1.1:** Although the paper highlights KL-projection perspective as a key contribution, I think the contribution is quite weak because first this normalization issues have been shown by previous works and eq(5) holds but is just another way to express the same observations mathematically.
>
>
> **Response to Weakness 1.1:** Thank you for this thoughtful feedback. We note that the observation that structured generations can degrade generation accuracy is not new[1,2], and we **do not claim** that phenomenon itself as a contribution. Our contributions are more specifically:
>
> - We propose **Draft-Conditioned Constrained Decoding (DCCD)**, a simple training-free two-stage inference method that first generates an unconstrained draft and then applies constrained decoding conditioned on that draft.
>
> - We provide a **feasible-mass / KL-projection view of constrained decoding**, showing how repeated masking and renormalization induce a cumulative projection tax and trajectory-dependent distortion.
>
> - Across multiple structured reasoning benchmarks and constraint types, DCCD **improves strict structured accuracy by up to +24 percentage points** over standard constrained decoding, with especially large gains for smaller models
>
>
>
>
> > **Weakness  1.2:** On the empirical improvements, the improvements seem significant. However, as I raised in the question section, I am not able to see exactly how the few-shot constrained decoding is performed; I think the question is important because few-shots share the same motivation as the paper by making the structured prediction familiar to the LLM.
>
> **Response to Weakness 1.2:** Please refer to our Response to Question 1, where we clarify exactly how this baseline is constructed and what is shown to the model.
> - More importantly, even against this familiarity-based baseline, DCCD consistently performs better across model scales and datasets. For example, on GSM8K with the 1.5B model, Constrained Few-Shot achieves 48.22, while DCCD improves this to 73.92; on MATH500 with the 7B model, Constrained Few-Shot achieves 44.20, while DCCD improves this to 52.80. These comparisons are reported in Table 2 of the paper.
>
>
> > **Weakness  1.3:** Finally, the paper proposed "predict-and-refine" method needs to cite significant LLM works as such methology has been proposed in many other domains; very minorly, constrained decoding has been formulated and experimented with a CFG backbone earlier than what paper cited, see https://aclanthology.org/P16-1127.pdf and https://aclanthology.org/P17-1041.pdf.
>
> **Response to Weakness 1.3:** Thank you for these suggestions. We will acknowledge earlier work on CFG- and syntax-aware constrained generation, and we will add the cited ACL 2016 and ACL 2017 papers in the revised version. The ACL 2016 paper studies grammar-constrained sequence prediction for semantic parsing, and the ACL 2017 paper studies syntax-driven neural generation via grammar actions over ASTs.
>
> **At the same time, we note that our setting is different.** Those works focus on structured generation in semantic parsing or code generation with grammar-based action spaces, whereas our paper studies **modern token-masking constrained decoding** under hard output schemas/grammars for LLMs, analyzes its distortion through **feasible mass and KL projection**, and proposes a  **draft-conditioned two-stage inference procedure** motivated by that view. We will also broaden the related-work discussion to better position DCCD relative to prior staged or predict-then-refine style methods.
>
>
> References:
>
> [1] Tam, Zhi Rui, et al. "Let me speak freely? a study on the impact of format restrictions on performance of large language models." arXiv preprint arXiv:2408.02442 (2024).
> [2] Castillo, Dylan. 2024. “Structured Outputs Can Hurt the Performance of LLMs.” December 8, 2024.

---

> > ### Author Rebuttal · Reviewer_NV7k · 2026-04-04
> >
> > Thanks for authors for carefully answering all my concerns. If the authors implement all the changes as discussed, I am also fine to accept this paper (although the score is not changed).

---

> > > ### Author Response · Authors · 2026-04-04
> > >
> > > We thank the reviewer for the positive acknowledgment.
> > >
> > > We confirm that we are fully committed to incorporating all updates discussed in the rebuttal into the final version, (i) including clarifying the few-shot baseline presentation in Appendix G, (ii) adding the suggested prior citations, and (iii) expanding the related-work discussion accordingly.
> > >
> > > Since the concerns are now fully resolved, we would be grateful if you would consider updating your score to reflect your assessment.

---

### Official Review · Reviewer_Wnof · 2026-03-11

**Soundness:** 2
**Presentation:** 3
**Significance:** 2
**Originality:** 2
**Overall Recommendation:** 3
**Confidence:** 4

**Summary:**

This paper formalizes the distortion that standard constrained decoding (CD) introduces via token-level masking and renormalization, expressing per-step distortion as a closed-form function of feasible mass. It then proposes Draft-Conditioned Constrained Decoding (DCCD), a simple generation strategy that first generates an unconstrained draft, then applies constrained decoding conditioned on that draft. The paper argues that conditioning increases feasible mass, reducing the cumulative “projection tax.” Evaluated on GSM8K, MATH500, GSM-Symbolic, and FOLIO across six models, DCCD shows consistent gains over standard CD. The paper also claims parameter efficiency and test-time scaling benefits.

**Compliance With Llm Reviewing Policy:**

Affirmed.

**Final Justification:**

Thanks for the additional experiments and the various clarifications. However, I cannot recommend acceptance given the quick additional experiment on a small subset of the benchmarks.
For the work to be publishable, the new analysis has to be more thorough

**Key Questions For Authors:**

1. Can you include an unconstrained baseline (no schema instructions, no constrained decoding) in Table 2? This would show how much accuracy is lost to structured generation overall and how much of that gap DCCD recovers.

2. Can you provide evidence that the feasible-mass mechanism, rather than the unconstrained draft stage alone, explains DCCD’s gains? For example, do gains correlate with measured feasible-mass improvements on real benchmark runs?

3. What is the relationship between draft correctness and final output correctness? When the draft is right, how often does Stage 2 preserve it, and when the draft is wrong, how often does Stage 2 recover? Separately, do gains vary with grammar tightness, as the feasible-mass theory would predict?

**Limitations:**

No. The paper includes an Impact Statement but does not discuss limitations. Missing discussions include: failure modes of the two-stage pipeline, the additional inference cost, and the relationship to thinking tokens.

**Strengths And Weaknesses:**

Strenghts
- S1: Clean formalization of CD distortion. Section 3.1 provides a closed-form characterization of per-step distortion (Eq. 5) and sequence-level reweighting (Eq. 8). Prior work noted that CD distorts output quality but did not quantify it this explicitly.
- S2: Consistent empirical improvement over CD. DCCD outperforms standard constrained decoding in all 24 model-dataset cells in Table 2, with especially large gains at smaller model scales. The evaluation spans multiple constraint types (JSON schemas, expression grammars, FOL) and is thorough across models and benchmarks.
- S3: Simple and practical. DCCD is easy to understand, training-free, and preserves syntax validity guarantees by applying the decoding time constraint in stage 2. It formalizes a decoupled generation pattern that exists as folk wisdom but has not been systematically evaluated for structured generation.
- S4: Clear presentation. The paper is well-written and well-structured. Algorithm 1 makes the method easy to reproduce, the toy example effectively builds intuition for the feasible-mass problem, and the progression from diagnosis to method to experiments is logical.

Weaknesses
- W1: The causal mechanism is asserted, not tested. The introduction presents per-token masking distortion as “the root cause” of CD’s reasoning degradation (lines 55–66). The paper does not directly test whether this specific mechanism explains DCCD’s empirical gains, or whether the gains come from something simpler, such as avoiding distribution shift from combining reasoning and structural constraints in a single pass. The formalism may be descriptive rather than explanatory, and the gains may have nothing to do with feasible-mass dynamics.
- W2: Missing unconstrained baseline. Table 2 compares CP, CF, CD, and DCCD, but omits the baseline where no constraint is imposed (neither at the prompt nor at decoding time), the natural ceiling for reasoning ability. Without it, the reader cannot assess how much of the accuracy gap between unconstrained and constrained generation DCCD actually closes. DCCD’s stage 1 is unconstrained, but the final output still passes through CD in stage 2, so it is not a substitute for this baseline.
- W3: Insufficient analysis of when or why DCCD helps. The paper reports that DCCD improves over CD in all configurations and includes supplementary analyses (Section 5.3, Appendices D/J/K), but none sufficiently explain when or why the method works. Two analyses are missing: (1) the relationship between draft correctness and final output correctness, i.e., when the draft is right, how often does stage 2 preserve it, and when the draft is wrong, how often does stage 2 recover; (2) whether gains correlate with grammar tightness, since tighter grammars produce lower feasible mass, predicting larger DCCD gains. The authors have the formalism to test this but do not.
- W4: No connection to thinking tokens. Thinking tokens are now ubiquitous and share the same pattern as DCCD. The thinking tokens are unconstrained and the non-thinking tokens are structured, within a single generation pass. This is a clear gap in their analysis and a missed opportunity. If thinking tokens performed similarly to DCCD, then the KL-projection view from Section 3.1 could explain why thinking tokens help for structured generation, which would have strengthened the paper’s theoretical contribution.
- W5: DCCD-specific theory is thin. I appreciated Section 3.1’s formalism, but the DCCD-specific theory (Section 4.1) does not deliver on its framing. The paper presents “feasible mass is context-dependent” as a key insight, but Eqs. 10–11 restate the Section 3.1 diagnostic with an additional conditioning variable, without characterizing when or by how much feasible mass improves. The buildup promises a theoretical explanation for DCCD, but the payoff is a conceptual bridge.

---

> ### Author Rebuttal · Authors · 2026-03-31
>
> We thank you for your valuable feedback.
>
> ## Response to Questions
>
> **Response to Question 1:**
>
> Yes. In Table 2 of the paper, we reported only strict structured accuracy, where a prediction must be both correct and exactly structure-compliant. Under that metric, unconstrained decoding is not a structured-generation baseline, since it does not enforce the required structure.
>
> As the reviewer suggested, it is still an important semantic reference point. We therefore add unconstrained results on MATH500 in Table 7 below. Here, **unconstrained decoding is evaluated with standard task accuracy, while CD and DCCD are evaluated with strict structured accuracy.** This shows both (i) the overall gap introduced by structured generation and (ii) how much of that gap DCCD recovers relative to CD.
>
> **Table 7: Unconstrained, CD, and DCCD on MATH500.**
> |Model|Unconstrained|CD|DCCD|Gain over CD|
> |---|---:|---:|---:|---:|
> |1B|16.40|5.20|13.80|+8.60|
> |1.5B|40.00|14.80|34.60|+19.80|
> |3B|51.60|31.40|44.20|+12.80|
> |7B|68.00|37.80|53.60|+15.80|
>
> **Response to Question 2:**
>
> Our claim is not that the unconstrained draft stage alone explains the gains. Rather, the **key mechanism** is that conditioning on the draft increases the feasible mass seen by Stage 2, which reduces the projection tax and makes the final structured realization less distortive. Table 5 ([please refer to Response to Weakness 2 of Reviewer MXdn](https://openreview.net/forum?id=hWDUpdwRZp&noteId=f2hcuUQrnE)). shows that models with larger feasible-mass improvement from CD to DCCD also tend to obtain larger accuracy gains on MATH500. Quantitatively, the **correlation coefficient is 0.71**, indicating a positive and statistically significant association.
>
> **Response to Question 3:**
>
>  We agree this is an important breakdown. In Table 8, we therefore analyze the relationship between draft correctness and final output correctness by separating cases where the Stage-1 draft is correct versus incorrect. This lets us measure how often Stage 2 preserves a correct draft and how often it can recover from an incorrect one.
>
> **Table 8: Relationship between draft correctness and final output correctness.**
> |Model|Draft Correct → Final Correct|Draft Wrong → Final Correct|
> |---|---:|---:|
> |Llama-3.2-1B-Instruct|72.9|0.2|
> |Qwen2.5-1.5B-Instruct|78.5|3.3|
> |Qwen2.5-3B-Instruct|83.6|2.4|
> |Qwen2.5-7B-Instruct|75.7|1.3|
>
> **Grammar tightness:** Yes. In Table 9, we also test whether tighter output constraints lead to larger degradation, as the feasible-mass view predicts. On GSM8K, we compare a more flexible schema where the answer field is a string against a stricter schema where the answer field must be an integer. The stricter schema reduces CD accuracy in both cases, with the larger drop appearing for the 3B model.
>
> **Table 9: Effect of schema tightness on CD accuracy on GSM8K.**
> |Model|CD (Flexible schema: answer as string)|CD (Stricter schema: answer as int)|Drop|
> |---|---:|---:|---:|
> |Qwen2.5-1.5B-Instruct|49.36|45.94|3.42|
> |Qwen2.5-3B-Instruct|81.58|63.00|18.58|
>
> **This is consistent with the feasible-mass view:** tighter schemas make valid continuations more restrictive and can increase the distortion introduced by direct constrained decoding.
>
> ## Response to Weaknesses
>
> **Responses to Weaknesses 1–3:** Please refer to our **Responses to Questions 1–3**, which directly address these points.
>
> **Response to Weakness 4:**
>
> We agree this is an interesting connection. Thinking tokens and DCCD share the same high-level intuition: semantic reasoning happens before the final structured answer is produced. However, DCCD makes this separation explicit: it first generates an unconstrained draft and applies constrained decoding only in the second stage.
>
> To test this connection, Table 10 reports a simple comparison on one representative setting using a reasoning-capable 1.5B model on MATH500. The results show that adding constrained decoding only at answer generation degrades performance relative to unconstrained reasoning, while DCCD largely preserves it.
>
> **Table 10: Comparing thinking-style generation with DCCD on MATH500.**
> | Model | Dataset  | Unstructured Answer  | Thinking + CD at Answer Stage | DCCD (R1 1.5B + Qwen 2.5 1.5B) |
> |---|---|---|---|---|
> | Deepseek R1 Qwen 1.5B  | MATH500 |  63.2 | 51.2 | 62.6  |
>
> **Response to Weakness 5:**
>
> We note that the DCCD theory is a mechanism-based explanation rather than a full predictive theory. **Our intended contribution in Section 4.1 is to show why draft conditioning can help:** by changing the context seen by Stage 2, it can increase feasible mass and thereby reduce the projection tax introduced by constrained decoding. We do not claim a theorem characterizing the exact amount of improvement for arbitrary drafts. To make the mechanism more concrete, **please refer to our Response to Question 2**, where we add empirical evidence relating feasible-mass improvement to final structured correctness.

---

> > ### Author Rebuttal · Reviewer_Wnof · 2026-04-03
> >
> > I thank the authors for a thorough rebuttal. Table 7 (unconstrained baseline) and Table 8 (draft correctness breakdown) are useful additions.
> > My central concern (W1) remains unaddressed. The paper claims that feasible-mass distortion is the root cause of CD's reasoning degradation. The rebuttal offers a correlation between feasible-mass improvement and accuracy gains, but correlation does not test the causal claim.
> > A straightforward test would compare accuracy under the same prompt (mentioning the structural constraint) with and without CD. Without CD, no tokens are masked and no decoder distortion occurs. If accuracy degrades similarly in both conditions, then the feasible-mass theory describes a secondary effect rather than the root cause. I strongly suspect that most of the degradation occurs without CD, with only a marginal additional cost from CD. Note that both conditions should be scored on answer correctness only, not strict structured accuracy.
> > This matters because the paper's contribution over prior practice depends on the theory. Draft-then-constrain is a known technique; the paper's novelty is the feasible-mass explanation of why it works. If the theory attributes gains to the wrong mechanism, what remains is an evaluation of an established approach.

---

> > > ### Author Response · Authors · 2026-04-03
> > >
> > > **Response:** We thank the reviewer for the concrete suggestion and for highlighting this important distinction.
> > >
> > > To directly address this concern, we include a **prompt-vs-CD** experiment to isolate whether the degradation comes primarily from merely mentioning the structural constraint in the prompt, or from applying **constrained decoding (CD)** itself. In all cases below, we follow the reviewer’s request and **evaluate answer correctness** only, rather than str
> > >
> > > Specifically, we evaluate three settings:
> > >
> > > - **A (plain prompt, No CD)** (as a baseline),
> > > - **B (prompt (with mentioning the structural constraint), No CD)**,
> > > - **C (prompt (with mentioning the structural constraint), with CD)**.
> > >
> > > This decomposition is important to measure the accuracy degradation gap from the baseline A.
> > > We present our results as follows:
> > >
> > > **Table 1.1: MATH500 (answer accuracy only) where negative numbers in bracket shows accuracy drop.**
> > > | Model Size | A (plain prompt, no CD) | B (prompt with structural constraint, no CD) | C (prompt with structural constraint, with CD) |
> > > |:-----------|------------------------:|---------------------------------------------:|-----------------------------------------------:|
> > > | Llama 3.2 1B Instruct | 24.6 | 20.6 (-4.0) | 6.0 (**-18.6**) |
> > > | Qwen 2.5 1.5B Instruct | 45.6 | 41.4 (-4.2) | 15.0 (**-30.6**) |
> > > | Qwen 2.5 3B Instruct | 54.2 | 50.3 (-3.9) | 33.4 (**-20.8**) |
> > > | Qwen 2.5 7B Instruct | 71.4 | 71.2 (-0.2) | 43.6 (**-27.8**) |
> > > | Llama 3.1 8B Instruct | 39.2 | 38.0 (-1.2) | 28.6 (**-10.6**) |
> > > | Qwen 2.5 14B Instruct | 76.0 | 70.8 (-5.2) | 47.6 (**-28.4**) |
> > >
> > >
> > > ### **Key Takeaway: Degradation is substantially larger once CD is applied (consistent with our claim in the paper)**.
> > >
> > > ### **Remarks**
> > >
> > > Importantly, in all cases the degradation from **A to B is small**, while the additional degradation from **B to C is much larger**. This indicates that the major degradation appears when masking is applied, rather than when the structural constraint is merely mentioned in the prompt.
> > >
> > > These results directly address the reviewer’s concern. The degradation does not occur primarily from merely mentioning the structural constraint in the prompt. Rather, the major loss appears when decoder-time masking is introduced. In other words, prompt-level structural awareness explains only a small fraction of the observed degradation in most settings, while the much larger additional drop arises from constrained decoding itself.
> > >
> > > We thank the reviewer for suggesting this experiment and insights. We will add this detailed analysis in the final version of the paper.

---

### Official Review · Reviewer_MXdn · 2026-03-12

**Soundness:** 3
**Presentation:** 2
**Significance:** 2
**Originality:** 3
**Overall Recommendation:** 3
**Confidence:** 3

**Summary:**

This paper addresses the degradation of semantic reasoning capabilities in large language models (LLMs) caused by standard constrained decoding (CD) during structured generation tasks (e.g., JSON outputs, API calls). The authors propose "Draft-Conditioned Constrained Decoding" (DCCD), a training-free, two-step inference procedure: it first generates an unconstrained semantic draft, and then applies hard-constrained decoding conditioned on this draft. This approach guarantees 100% structural validity without severely distorting the model's original semantic distribution.

**Compliance With Llm Reviewing Policy:**

Affirmed.

**Key Questions For Authors:**

1. Could you provide an ablation study comparing the $S^{(k)}$-based selection against Majority Voting and an LLM-as-a-judge approach? Why should we assume that maximizing the structural conversion probability aligns with maximizing reasoning accuracy?
2. In the CD baseline, is the length and freedom of the internal reasoning artificially truncated by XGrammar's masking mechanism? If you force Stage 1 of DCCD to be extremely brief, does the performance drop precipitously?

**Limitations:**

The paper offers highly insightful theoretical explanations (via the KL-projection tax lens) for why structured generation harms LLM reasoning. The proposed Draft-Conditioned approach, while fundamentally a low-level implementation of a two-stage prompt chain, is algorithmically elegant and heavily supported by empirical data on smaller models. However, the claims regarding "parameter efficiency" downplay the severe latency costs introduced by dual-pass autoregression, and the sorting heuristic for $K>1$ lacks theoretical consistency.

**Strengths And Weaknesses:**

Strengths:
1. The paper targets a critical bottleneck in current agentic workflows and tool-calling frameworks: guaranteeing format validity often comes at the expense of reasoning capability. The Toy Example in Figures 2 and 3 intuitively and profoundly illustrates the root cause of this phenomenon (i.e., error accumulation driven by low-probability formatting tokens in early generation steps).
2. The derivation from Eq. 5 to Eq. 8 is theoretically sound. Explaining the introduction of an auxiliary context $d$ (the draft) as a mechanism to increase the feasible mass $\tilde{\alpha}(\tilde{h}_t)$ provides a solid probabilistic grounding for the human-like heuristic of "drafting before formatting."

Weaknesses:
1. In the standard Constrained Decoding (CD) baseline, the model is strictly constrained from the very first token. In contrast, DCCD's Stage 1 operates completely unconstrained, effectively executing an implicit, unbounded Chain-of-Thought (CoT). The experiments do not clarify whether the CD baseline was allocated an strictly equal token compute budget within its JSON steps field. If DCCD generates an average of 300 reasoning tokens in its draft, but CD only manages to generate 100 tokens under strict formatting pressure, the performance gain may simply stem from "generating more reasoning tokens" rather than "avoiding the KL-projection tax."
2. In the multi-draft sampling scenario ($K>1$), Algorithm 1 (Line 12) utilizes the cumulative log feasible mass $S^{(k)} = \sum \log(\tilde{\alpha}_t^{(k)})$ as the criterion for selecting the best draft. This metric solely measures "format alignment" (i.e., how easily a draft can be forced into the target structure without heavy renormalization) rather than "reasoning correctness." A confidently incorrect draft could similarly yield a very high $\tilde{\alpha}_t$. The paper lacks mathematical proofs or thorough ablation studies demonstrating that "smoothness of format conversion" positively correlates with "mathematical/logical correctness."

---

> ### Author Rebuttal · Authors · 2026-03-31
>
> We thank you for your valuable feedback.
> ## Response to Weaknesses
>
> **Response to Weakness 1:**
>
> Thank you for this important point. Our current experiments are not intended to give DCCD a privileged reasoning channel. In our structured reasoning tasks, the target schema itself contains intermediate reasoning fields (e.g., JSON steps), so the CD baseline is also allowed to generate reasoning tokens.
>
> - **The key difference is how reasoning is generated:** CD must produce those reasoning tokens under token-level masking and renormalization from the very first token, whereas DCCD first forms a semantic draft unconstrained and only then performs structured realization.
>
> To address this directly, we add a budget-matched ablation in Table 3, where CD and DCCD are compared under the same total generation budget, counting both Stage 1 and Stage 2 tokens for DCCD.
>
> **Table 3: Budget-matched comparison between CD and DCCD on MATH500.** For the DCCD, we split the total budget equally to stage 1 and stage 2.
> | Model  | Total Budget | CD Accuracy(%) | DCCD Accuracy(%) |
> |---|---:|---:|---:|
> | 1B  | 900 | 6.40 | 15.8 |
> | 1.5 B | 600 | 14.00 | 17.4 |
> |8B  | 900 | 28.60 | 32.2 |
>
> We also add a draft-length ablation that caps the Stage-1 draft length at fixed values. As shown in Table 4 below, DCCD retains an advantage even when the draft is restricted.
>
> **Table 4: DCCD draft-length ablation on MATH500.**
> | Model  | CD    | DCCD (256) | DCCD (512) | DCCD (1024) | DCCD (2048) |
> |------|-------|------------|------------|-------------|-------------|
> |8B | 28.60 | 29.20| 30.50| 34.80| 35.00|
>
> It is important to note that, since we are generating a draft, these results suggest that the gain is not explained solely by a larger reasoning budget.
>
> **Response to Weakness 2:**
>
> Thank you for this thoughtful point. We note that the cumulative log feasible mass used in Algorithm 1 is not a direct measure of reasoning correctness. **Its role is different:** it measures how much constraint-induced distortion the second-stage incurs when converting a draft into a valid structured output. This is also why the paper presents feasible-mass scoring only as one practical instantiation of the broader DCCD framework, and explicitly notes that (Line 256, right column) alternative late-selection criteria such as verifier scores, judges, or majority voting can also be used.
>
> Our intended claim is therefore more precise: **higher cumulative feasible mass should correspond to smaller projection tax in Stage 2, which makes the structured realization less distortive and therefore more likely to preserve the semantic content of the draft.**
>
> **Cumulative log feasible mass and final structured correctness:** We add an additional ablation that measures the empirical relationship between cumulative log feasible mass and final structured correctness across sampled drafts. Presented in Table 5 below.  **Quantitatively, the correlation coefficient is 0.71, indicating a positive and statistically significant association.**
>
> **Table 5: Feasible mass and structured accuracy of constrained decoding (CD) and DCCD on MATH500 across models.**
> |Model|Accuracy (CD)|Accuracy (DCCD)|Feasible Mass (CD)|Feasible Mass (DCCD)|
> |---|---:|---:|---:|---:|
> |Llama-3.2-1B-Instruct|0.0660|0.1260|0.4563|0.8181|
> |Qwen2.5-1.5B-Instruct|0.1300|0.3260|0.6690|0.7821|
> |Qwen2.5-3B-Instruct|0.3160|0.4300|0.6953|0.7425|
> |Qwen2.5-7B-Instruct|0.4140|0.5280|0.8019|0.9156|
> |Llama-3.1-8B-Instruct|0.2740|0.3340|0.7144|0.8822|
> |Qwen2.5-14B-Instruct|0.4600|0.5720|0.7901|0.8729|
>
> ## Response to Questions
>
> **Response to Question 1:**
>
> We agree that this comparison is useful. Feasible mass is not a direct correctness score; it is a proxy for how little distortion the constrained projection step introduces. We therefore compare feasible-mass selection with majority voting in Table 6 below, using the same sampled drafts.
>
> **Table 6: Comparison of multi-draft selection rules on MATH500 with K=5.**
> |Model|Feasible-Mass Sel. Acc. (%)|Majority Voting Acc. (%)|
> |---|---:|---:|
> |3B|47.0|50.0|
> |8B|35.2|39.4|
>
> **Response to Question 2:**
>
> We note that in our CD baseline, the model **is allowed to produce intermediate reasoning.** For GSM8K/MATH500, the constrained output itself contains a steps field, and the Appendix case study J.2.1, J.2.2 shows CD generating multiple reasoning steps inside the structured output. The difference is not that CD is prevented from reasoning, but that it must reason under hard masking from the first token.
>
> As suggested by the reviewer, we also evaluated brief Stage-1 drafts (shown in Table 4 above). Performance drops as the draft is made shorter, but not precipitously.

---

### Official Review · Reviewer_sgod · 2026-03-12

**Soundness:** 3
**Presentation:** 2
**Significance:** 3
**Originality:** 3
**Overall Recommendation:** 4
**Confidence:** 2

**Summary:**

The paper presents DCCD, a mechanism to boost performance in constrained decoding. Modern LLM applications often require structured output, such as jsonl output. However, DCCD argues that using constrained decoding will distort the model’s probability distribution because valid tokens may have very low probability, leading to poor generation quality. The paper then proposed that Instead of forcing reasoning inside constraints, it adopts a two step generation, with the first round doing free style generation and the second round still using constrained decoding. But now the model sees the reasoning context which makes the formatting tokens are more expected, leading to better generation. Empirical analysis shows improved performance with DCCD comparing with vanilla constrained decoding.

**Compliance With Llm Reviewing Policy:**

Affirmed.

**Final Justification:**

The rebuttal looks good to me, and I don’t have more questions — I am already giving positive score.

**Key Questions For Authors:**

N/A

**Limitations:**

Yes

**Strengths And Weaknesses:**

Strengths:

The phenomenon observed in this paper is interesting, that by adding up the free form generation before constrained decoding, it can substantially improve the logits distribution and generation accuracy.

There is mathematical derivation in this paper to explain the observed improvements.

Experiments demonstrate consistent gains over standard constrained decoding.

Weaknesses

This is not a major concern as the paper is mainly accuracy oriented, but the proposed method seems to decrease the inference efficiency, that each generation requires at least two end-to-end inference pass for getting the final answer. This could not suit in latency sensitive scenarios.

If I understand it correctly, the effectiveness of the approach shall depend on the quality of the generated draft, as the draft directly shapes the probability distribution used to generate the structured output. So the method works best when the draft captures correct semantic reasoning, because the second stage largely performs structured realization of what the draft suggests. It could be beneficial to discuss more regarding on how draft errors affect the final constrained output.

---

> ### Author Rebuttal · Authors · 2026-03-31
>
> We thank you for your valuable feedback.
>
> > **Weakness 1:** This is not a major concern as the paper is mainly accuracy oriented, but the proposed method seems to decrease the inference efficiency, that each generation requires at least two end-to-end inference pass for getting the final answer. This could not suit in latency sensitive scenarios.
>
> **Response to Weakness 1:** This is an interesting point. Our main clarification is that, in **structured generation**, latency should be interpreted jointly with output validity and task accuracy. A fast method that does not reliably satisfy the required schema or grammar is not directly usable in practice. Thus, the relevant comparison is not only “how many passes are used,” but rather the overall tradeoff among accuracy, adherence, and end-to-end runtime.
>
> To make this explicit, we added the comparison below on MATH500, measured with the vLLM inference engine in batched mode on an NVIDIA H100. DCCD is slower than direct constrained decoding by a modest margin, but consistently achieves much higher structured accuracy while maintaining very high adherence.
>
> **Table 1: Accuracy, adherence, and total wall-clock time on MATH500.**
>
> |Model|Method|Accuracy ↑|Adherence ↑|Time (min) ↓|
> |---|---|---:|---:|---:|
> |Llama-3.2-1B|Unconstrained|16.40|0.00|0.294|
> ||Constrained|5.20|90.80|0.999|
> ||DCCD (ours)|13.80|96.80|1.358|
> |Qwen2.5-1.5B|Unconstrained|40.00|0.00|0.376|
> ||Constrained|14.80|96.40|0.892|
> ||DCCD (ours)|34.60|98.60|1.298|
> |Qwen2.5-3B|Unconstrained|51.60|0.00|0.205|
> ||Constrained|31.40|96.20|1.190|
> ||DCCD (ours)|44.20|98.40|1.445|
> |Qwen2.5-7B|Unconstrained|68.00|0.00|0.148|
> ||Constrained|37.80|98.00|1.573|
> ||DCCD (ours)|53.60|100.00|1.781|
>
>
> > **Weakness 2:** If I understand it correctly, the effectiveness of the approach shall depend on the quality of the generated draft, as the draft directly shapes the probability distribution used to generate the structured output. So the method works best when the draft captures correct semantic reasoning, because the second stage largely performs structured realization of what the draft suggests. It could be beneficial to discuss more regarding how draft errors affect the final constrained output
>
>
> **Response to Weakness 2:** We agree that the quality of the stage-1 draft is an important factor in DCCD. Since the second stage conditions on the draft, a better draft generally leads to a better structured output, while errors in the draft can limit the final result.
>
> We examined this directly on MATH500, where the unconstrained draft itself can be evaluated. The results show a consistent pattern across model sizes: unconstrained draft accuracy is highest, standard constrained decoding degrades sharply, and DCCD recovers a substantial fraction of that loss. For example, on MATH500 with the 14B model, unconstrained decoding achieves 76.0, constrained decoding drops to 47.6, and DCCD improves this to 58.6.
>
> **Table 2: MATH500 Accuracy comparison.** The unconstrained draft is included only to measure the semantic quality of stage 1.
>
> | Model Size | Unconstrained Draft  ↑ | CD ↑| DCCD (ours) ↑|
> |---|---:|---:|---:|
> | Llama 3.2 1B Instruct  | 24.6 | 6.0  | 19.8 |
> |  Qwen 2.5 1.5B Instruct | 45.6 | 15.0 | 38.2 |
> | Qwen 2.5 3B  Instruct | 54.2 | 33.4 | 46.8 |
> | Qwen 2.5 7B  Instruct | 71.4 | 43.6 | 52.8 |
> | Llama 3.1 8B  Instruct | 39.2 | 28.6 | 35.0 |
> | Qwen 2.5 14B Instruct | 76.0 | 47.6 | 58.6 |
>
>
> **This supports two conclusions:**
>
> (1) draft quality matters because DCCD is bounded by the semantic quality of the first-stage draft; and
> (2) DCCD is still useful even when the draft is imperfect, because it consistently recovers much of the loss introduced by direct constrained decoding.
>
> We will add this discussion, along with the diagnostic comparison below, to the revised paper.

---

> > ### Author Rebuttal · Reviewer_sgod · 2026-04-02
> >
> > I am good with the response. I will maintain my positive score here.

---

> > > ### Author Response · Authors · 2026-04-02
> > >
> > > Dear Reviewer,
> > >
> > > Thank you for your time and for acknowledging the rebuttal. We are happy that the rebuttal addressed your concerns. Thank you for your positive scores.
> > >
> > > Regards,
> > >
> > > Authors

---

### Decision · Program_Chairs · 2026-04-30

**Decision:**

Accept (regular)

**Comment:**

The paper proposes a simple, training-free two-stage decoding strategy (DCCD) that is practically relevant and consistently improves structured generation accuracy over constrained decoding, with solid empirical support and a clear formulation of the distortion issue . While several reviewers raise valid concerns about novelty, causal claims, and incomplete analysis, the rebuttal addresses most practical issues and adds useful ablations. The remaining weaknesses mainly affect theoretical positioning rather than empirical usefulness. Overall, this is a solid and timely contribution with clear practical impact, though not without limitations.